# A stochastic compartmental model to simulate intra- and inter-species influenza transmission in an indoor swine farm

**Eric Kontowicz[1]\*, Max Moreno-Madriñan[2,3], Darryl Ragland[4], Wendy Beauvais[1]**

**1** Department of Comparative Pathobiology, Purdue University College of Veterinary Medicine, West Lafayette, Indiana, **2** Global Health Program, DePauw University, Greencastle, Indiana, **3** Department of Global Health, Indiana University, Indianapolis, Indiana, **4** Department of Veterinary Clinical Sciences, Purdue University College of Veterinary Medicine, West Lafayette, Indiana

\* ekontowi@purdue.edu

**Data Availability Statement:** Code and data used for this manuscript are available at https://github.com/ekontowicz/Compartmental-model-to-

## Abstract

Common in swine production worldwide, influenza causes significant clinical disease and potential transmission to the workforce. Swine vaccines are not universally used in swine production, due to their limited efficacy because of continuously evolving influenza viruses. We evaluated the effects of vaccination, isolation of infected pigs, and changes to workforce routine (ensuring workers moved from younger pig batches to older pig batches). A Susceptible-Exposed-Infected-Recovered model was used to simulate stochastic influenza transmission during a single production cycle on an indoor hog growing unit containing 4000 pigs and two workers. The absence of control practices resulted in 3,957 pigs [0–3971] being infected and a 0.61 probability of workforce infection. Assuming incoming pigs had maternal-derived antibodies (MDAs), but no control measures were applied, the total number of infected pigs reduced to 1 [0–3958] and the probability of workforce infection was 0.25. Mass vaccination (40% efficacious) of incoming pigs also reduced the total number of infected pigs to 2362 [0–2374] or 0 [0–2364] in pigs assumed to not have MDAs and have MDAs, respectively. Changing the worker routine by starting with younger to older pig batches, reduced the number of infected pigs to 996 [0–1977] and the probability of workforce infection (0.22) in pigs without MDAs. In pigs with MDAs the total number of infected pigs was reduced to 0 [0–994] and the probability of workforce infection was 0.06. All other control practices alone, showed little improvement in reducing total infected pigs and the probability of workforce infection. Combining all control strategies reduced the total number of infected pigs to 0 or 1 with a minimal probability of workforce infection (<0.0002–0.01). These findings suggest that non-pharmaceutical interventions can reduce the impact of influenza on swine production and workers when efficacious vaccines are unavailable.

## Introduction

Influenza A viruses (IAV), members of the Orthomyxoviridae family, are common respiratory viruses that infect humans and animals, and have a worldwide distribution [1]. Domestic pigs

simulate-interspecies-influenza-transmission-in-an-indoor-hog-grower-unit.

**Funding:** MM, DR, and WB were awarded grant from the National Pork Board (https://www.porkcheckoff.org). The grant number is NFB#21-100. All the authors received salary support on the same grant (NFB#21-100). The funders had no role in study design, data collection and analysis, decision to publish, or preparation of the manuscript.

**Competing interests:** The authors have declared that no competing interests exist.

(*Sus domesticus*) play a role in the maintenance of influenza viruses globally (especially subtypes H1N1, H3N2, and H1N2 within the United States) [2, 3]. Influenza viruses associated with swine can spill over to humans or other species. In most cases, the virus is not well adapted to transmission via the new host, however host adaptation can evolve via point mutations and genetic reassortment. Further, transmission within and across different host species can contribute to viral evolution leading to more virulent or pathogenic strains [4]. The risk that this process leads to a pandemic is difficult to quantify but is non-negligible [5–7]. In 2009 a novel subtype of H1N1 IAV spilled over from swine to humans leading to a pandemic which was estimated to infect approximately 60.8 million Americans [8], and one to three billion people worldwide (15–45% of the world's population) [9]. Swine and avian hosts played a key role in the 1918 influenza pandemic, which killed 50–100 million people in a time when the world population was 1.7 billion [10]. The swine-associated component of the 2009 H1N1 pandemic viral genome closely resembles 1918 H1N1 and sera from the survivors of the 1918 pandemic can neutralize 2009 H1N1 [11]. This demonstrates the key role swine play in the evolution of influenza.

In both pigs and humans, influenza viruses are mainly transmitted via inhalation of fomites and aerosols that are generated through breathing, coughing or sneezing [12]. Additionally, indirect contact with fomites on contaminated equipment or worker clothing have also been implicated as routes of transmission for IAV in swine populations [12]. On typical US swine farms, morbidity can be extremely high (up to 100% of the population) while mortality is typically low (< 1%) [6]. However, in naïve populations, mortalities can exceed 10% of the infected population [13]. Infected pigs are estimated to lose 5 to 12 pounds in body weight over a three-to-four-week period [14–16], leading to prolonged finishing times which contribute to the economic burden that influenza places on swine producers.

In addition to economic impacts, swine-associated influenza may also be transmitted to the swine workforce. One study found swine workers' odds of elevated antibodies to swine H1N1 virus (A/swine/WI/238/97) to be 54.9 (95% confidence interval [95% CI]13.0–232.6) times those of non-swine industry worker controls [17]. In a separate seroprevalence study evaluating farmers, meat processing workers (from pork production facilities), veterinarians with swine exposure and controls with no known contact with swine, it was found that farmers had the highest odds of swine influenza exposure (Odds Ratio [OR], 35.3; 95% CI, 7.7–161.8) [18]. Another study using whole genome sequencing found that IAV samples from five workers had identical clade classification to swine-origin IAV samples from pigs on the same farm [19]. Evidence also suggests that IAV can be transmitted from workers to pigs. One study in the Czech Republic found the presence of antibodies in pigs to human-origin influenza [20]. In the US, transmission of 2009 H1N1 influenza from workers back into the pig population has frequently been recorded [21]. Some influenza viruses (e.g., A/California/VRDL101/2009/H1N1) transmitted from humans to pigs have not been shown to be transmitted within pig populations while other influenza viruses (e.g. A/swine/Illinois/A01395201/2013/H1N1) transmitted by humans can become well established in pigs as determined via phylogenic relationships [21]. Reverse zoonotic transmission (human-to-animal transmission) of influenza virus leading to sustained swine-to-swine transmission have been reported relatively infrequently; 20 such events were reported between 1965 and 2013 [20]. However, it is highly likely that transmission events have gone undetected.

## Control measures

Currently, vaccination is one of the most practical and effective means to control and prevent influenza in pigs and humans alike. However vaccine efficacy is frequently suboptimal and

limited by the rapid evolution of influenza viruses [22, 23]. Whole, inactivated viruses with adjuvant are typically administered via intramuscular injection to either growing pigs or sows [24, 25]. Maternally-derived antibodies (MDAs) have been shown to protect offspring for approximately 6 weeks post-weaning [26]. Of the commercial inactivated vaccines used in the US, it is estimated that around 50% are autogenous [23] *i.e.* subtypes isolated from swine farms are used directly to make a vaccine for use in the same population [27]. In the US, veterinarians may request authorization to use these autogenous vaccines from the Food and Drug Administration. In addition to inactivated vaccines, viral vector vaccines, DNA vaccines and subunit IAV vaccines are commercially available but used less frequently [23, 27]. Reduced viral shedding due to vaccination has been documented in both humans [28] and pigs [29–31].

In addition to vaccination, grower farms may practice all-in all-out practices (with thorough cleaning and disinfection between batches), biosecurity practices such as limiting personnel and showering in and out, purchasing piglets from known sources, surveillance of incoming animals, isolation and quarantine [12, 32]. Ventilation systems have been shown to reduce the introduction of and transmission of pathogens across different rooms on farms [33–36]. Lastly, improved personal protective equipment (PPE) or adherence to PPE mandates can help reduce influenza transmission between hogs and workers. Despite these available control measures, IAV remains prevalent in US grower herds [37]. One study of an animal operation in Minnesota found that specialized footwear was the most commonly used form of PPE by workers and they were unlikely to use masks or gloves [38]. This study also found that handwashing before and after animal handling was not widely encouraged [38].

Etbaigha et al. (2018) [39] adapted a deterministic model first published by Reynolds et al. (2014) [40] to show that piglets play a key role in the maintenance of IAV on farrow-to-finish farms. Reynolds et al. further showed that implementing either homologous or heterologous mass vaccination was not sufficient to eliminate IAV on a wean-to-finish farm [40]. Using a stochastic approach, both Pitzer et al. [41] and White et al. [42] indicated that piglets are a key reservoir for IAV and vaccination alone will not eliminate the presence of IAV on breeding and finishing farms. Further, White et al. [42] showed that a combination of vaccination and biosecurity practices can reduce the likelihood of maintaining an endemic reservoir in piglets on a breeding farm. There is also a large body of literature that pertains to modeling influenza in populations and speices other than indoor swine production systems, that is beyond the scope of this review.

The goal of this study was to quantify the effectiveness of IAV control measures targeting both hogs and the workforce within a single, typical U.S. indoor hog-growing unit. The current model differs from previous research by: (1) incorporating stochastic transmission between pigs and the workforce, (2) evaluating multiple combinations of prevention, biosecurity and workplace policies on IAV dynamics within a US indoor growing unit, and (3) comparing the effectiveness of these measures in terms of cases averted in both pigs and the workforce.

## Methods

### Overview

We simulated stochastic transmission of a single influenza virus subtype amongst swine and the workforce during a single production cycle (approximately a six-month period). We developed our own models that closely resemble the compartmental modelling approach used by Pitzer et al [41] and White et al [42]. Our model is adapted to the swine population on a typical US hog growing production unit using an all-in, all-out approach and included interspecies

transmission between the workforce and swine. We first tested a baseline model that assumes current typical industry practices and no vaccination among pigs or workers. We then explored transmission dynamics under different intervention strategies to evaluate the change in the number of pigs infected, the probability of at least one workforce infection, and the time until interspecies transmission (to workers or pigs). Intervention strategies included mass vaccination (20% effectiveness, 40% effectiveness, or 60% effectiveness), isolation of infected pigs, changes to worker routine and improved PPE including masks. These strategies were evaluated under two separate conditions: 1. Incoming pigs were assumed to have no maternally derived antibodies (MDA), or 2. All incoming pigs were assumed to have MDAs for the first three weeks.

## Population dynamics

Two populations were simulated: the growing pig population and the human workforce on the farm (**Fig 1**). The farm layout, number of pigs per room, and total number of workers employed on the farm closely resemble a typical commercial hog grower unit in Indiana and other swine intensive areas in the Midwest. These details were determined via informal expert consultation.

Pigs enter the grower facility at three weeks of age after they have been weaned at a breeding and farrowing facility. When entering the grower unit, each batch of weaned pigs are placed into a single room and remain in this room for 23 weeks, when it is assumed that they have reached market weight. In model scenarios which included MDAs we assumed the presence of

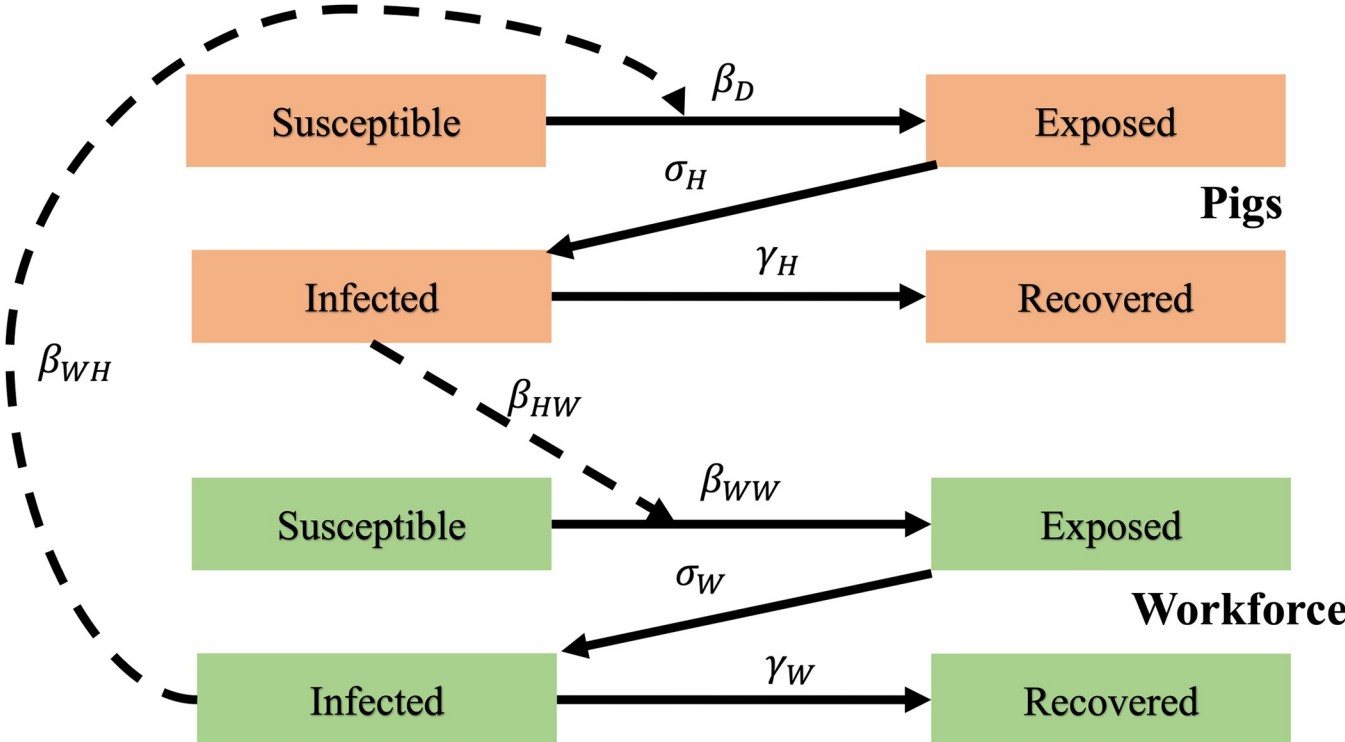

**Fig 1. Compartmental model to predict influenza transmission dynamics on a typical US hog grower unit.** We assume hogs entered at 3 weeks of age and remained for 23 weeks until they reached market weight. Each room houses 1,000 hogs and there is no direct contact between hogs in separate rooms, but indirect transmission can occur via aerosol and fomite transmission (e.g., contaminated equipment, worker clothes or boots). The workforce consists of two workers that visit each room daily and have an equal likelihood of contacting any hog.

MDAs for the first three weeks of the model [26]. We assumed an all-in / all-out system [43] where each room was filled two weeks apart to maximum capacity (1,000 hogs per room; 4,000 hogs in total). All pigs in each room leave the farm on the same day. For example, room 1 is filled to 1,000 pigs on day 1, room 2 filled on day 14, etc. There is no direct contact between pigs in separate rooms. It was assumed that only two workers were needed for husbandry tasks, such as feeding and performing daily checks [43, 44]. It was assumed that each worker had an equal probability of interacting with each hog.

## Transmission dynamics

The transmission of IAV on this growing farm is simulated using a Susceptible-Exposed-Infected-Recovered (SEIR) model. $S_i(t)$, $E_i(t)$, $I_i(t)$, and $R_i(t)$ are the number of susceptible, exposed, infected and recovered pigs respectively; $t$ is time in days, where $t \geq 0$, $i$ represents the room where the growing hog is housed, where $i$ is equal to 1, 2, 3 or 4, $j$ represents the adjacent rooms where other pigs are housed, where $j \in \{1,2,3,4\}$, $j \neq i$. Pigs become infected via within-room or between-room transmission. Between-room transmission occurs via air flow and movement of fomites such as contaminated workers' clothing, boots or equipment.

Here we describe the baseline model that assumes typical control measures are used. The SEIR models for the pigs are defined by the following set of differential equations:

**Pigs**

$$\frac{dS_i}{dt} = -\left((1-A)\left(\beta_D S_i I_i* - \beta_N S_i \left(\sum_{j=\min(j)}^{\max(j)} I_j\right)\right)\right) - \left(A\left(\beta_A S_i I_i* - \beta_M S_i \left(\sum_{j=\min(j)}^{\max(j)} I_j\right)\right)\right) - \beta_{WH} S_i I_W$$
$$- \mu S_i$$

$$\frac{dE_i}{dt} = \left((1-A)\left(\beta_D S_i I_i* + \beta_N S_i \left(\sum_{j=\min(j)}^{\max(j)} I_j\right)\right)\right) + \left(A\left(\beta_A S_i I_i* + \beta_M S_i \left(\sum_{j=\min(j)}^{\max(j)} I_j\right)\right)\right) + \beta_{WH} S_i I_W$$
$$- \sigma_H E_i - \mu E_i$$

$$\frac{dI_i}{dt} = \sigma_H E_i - \gamma_H I_i - \mu I_i$$

$$\frac{dR_i}{dt} = \gamma_H I_i - \mu R_i$$

$A$ represents a researcher dictated factor that is assumed to either be 1 or 0 to determine if MDAs are considered in the model or not. The models for the workforce use a similar structure to that of the pigs. Workers can infect each other ($\beta_{WW}$) or become infected from the pigs ($\beta_{HW}$). The parameters are described in **Table 1**. We assume no between-room transmission from swine to the workforce or between-room transmission between workforce members. We did not include a loss of immunity term in our model because we focused on single subtypes of influenza and single production cycles with a length of 23 weeks. Evidence suggests that natural infection with IAV can induce lasting protection against the same subtype [45]. The Gillespie Stochastic Simulation Algorithm direct method was used to model the random events of influenza transmission (intra- and inter-species), latency, and recovery for both pig and worker populations [46], using the most likely parameter values presented in **Table 1** as the mean transition rates. Distributions of parameter values were used for later sensitivity analysis [47].

**Table 1. Parameters for IAV SEIR model with description and values.**

| Parameter Description | Parameter Symbol | Most Likely Value (Parameters for Uncertainty Distribution) | Uncertainty Distribution (for sensitivity analysis) | Source |
|---|---|---|---|---|
| Within-room Transmission Rate | $\beta_D$ | 0.002 (0.0001–0.01) /day | Triangle | [29] |
| Within-room Transmission Rate with Maternal Derived Antibodies | $\beta_A$ | $\beta_D$ / 10 | Triangle | [29, 39] |
| Between-room Transmission Rate | $\beta_N$ | $\beta_d$/ 178 (0, $\beta_d$) | Uniform | [29, 48] |
| Between-room Transmission Rate with Maternal Derived Antibodies | $\beta_M$ | $\beta_A$/ 178 | Uniform | [29, 39] |
| Present Maternal Derived Antibodies Factor | A | 0, 1 | Fixed | NA |
| Workforce to Pig Transmission Rate | $\beta_{WH}$ | 0.00007 / day (0.000001, 0.0001) | Uniform | [49] |
| Pig to Workforce Transmission Rate | $\beta_{HW}$ | 0.00003 / day (0.000001, 0.0001) | Uniform | [50] |
| Isolated Pig to Workforce Transmission Rate | $\beta_{QHW}$ | 0.0 / day | Fixed | NA |
| Workforce to Isolated Pig Transmission Rate | $\beta_{WQH}$ | 0.0 / day | Fixed | NA |
| Workforce to Workforce Transmission Rate | $\beta_{WW}$ | 0.32 / day (0.0000001, 0.64) | Triangle | [51] |
| Latent Period for Pigs | $\sigma_H$ | 2 days (sd = 1) | Normal | [29] |
| Infectious Period for Pigs | $\gamma_H$ | 5 days (sd = 1) | Normal | [29] |
| Latent Period of Humans | $\sigma_W$ | 2 days (sd = 1) | Normal | [51] |
| Infectious Period for Humans | $\gamma_W$ | 3 days (sd = 1) | Normal | [51] |
| Natural Death Rate for Pigs | $\mu$ | 0.00028 / day | Fixed | [39] |
| Isolation Rate | q | 1, 0.5, 0.33 or 0.0 per infected hog / day | Fixed | NA |
| Pig Vaccine Efficacy | *Vacc Eff* | 80%, 40%, or 0% | Fixed | [52–54] |

Given that influenza is transmitted via the respiratory route in pigs and humans, we assumed density-dependent transmission for our models. We estimated that: $\beta_D = \frac{R_0}{N*D}$ [55], where $N$ is the number of pigs in a single room (1,000 hogs), $D$ is the duration of infectiousness (5 days) and $R_0 = 10$, based on experimental data using a triple reassortant H1N1 subtype (A/swine/IA/00239/04) [29]. Due to a paucity of empirical data quantifying interspecies transmission rates within indoor hog growing units, we estimated interspecies transmission rates based on simplifying assumptions. We estimated that

$\beta_{HW} = \frac{R_0}{N*D}$ [55], where $N = 4002$, $D = 5$ and $R_0 = 0.6$, based on an outbreak of influenza A (H3N2) among attendees of an agricultural fair [50]. We assumed that during this outbreak, a single pig was responsible for transmitting to the first three confirmed human cases. We estimated that $\beta_{WH} = \frac{R_0}{N*D}$, where $N = 4002$, $D = 3$ and $R_0 = 0.83$, based on an influenza (pH1N1) outbreak on a swine research farm in Alberta, Canada [49]. We assumed that the four infected workers were responsible for the first 10 swine influenza cases. In baseline simulations, we assumed that a single infected pig was introduced when the last room was filled on Day 42. Lastly, for simplicity, we assumed that infected pigs had the same mortality rate as non-infected pigs.

## Interventions

We tested four categories of interventions: mass swine vaccination, isolation of infectious pigs, a directional workforce routine, and improvement in worker PPE. For vaccination, we assumed that vaccine efficacy (%) corresponded to the proportion of the pig population that was completely protected from both infection and ongoing transmission (i.e. they started and remained in the recovered compartment) [56–59]. We assumed pigs were vaccinated the same day they were introduced to the farm (the day the room was filled) as this is common practice on most indoor pig farms of the US.

In order to simulate isolation, we incorporated four isolation compartments (one for each room) that could hold a maximum of 10 pigs (per isolation compartment). We assumed workers adhered strictly to PPE and N95 respirator use when working with these isolated animals and that these animals were visited only after all other pig rooms each day; therefore, we assumed no pig-to-human transmission and no between-room transmission from isolated pigs to other pigs. It was assumed that infectious pigs were moved to isolation pens at a rate of q (1 / mean days until the infectious pig is moved to isolation) until the isolation pen reached maximum capacity.

In order to simulate the directional workforce routine, we assumed that each worker always moved from the youngest pig cohort (the pigs that entered the farm most recently) to the oldest cohort of pigs. Under this assumption, the incidence attributed to between-room transmission is shown in **Table 2** (The baseline equations were modified accordingly.).

We also evaluated the impacts of improved PPE on reducing transmission from the workforce to pigs. We assumed workers wore either no facial covering, surgical masks, or N95 respirators and assumed perfect adherence. We assumed that the % reduction in viral particle inhalation due to masks corresponded to a similar reduction in the effective contact rates from workers to swine and/or from swine to workers [60]. While the this is an imperfect proxy, there is a general lack of data to allow for a more accurate estimate. Surgical masks are 68% effective [60], and designed to trap secretions from the wearer and therefore reduce transmission from the individual, and were assumed to reduce only $\beta_{WH}$ [61]. N95 masks are estimated to be 91% effective [60] and were assumed to reduce $\beta_{WH}$ and $\beta_{QHW}$ [61]. Like the models testing the effects pig-specific control measures an infected individual was introduced on day 42, when all rooms were at capacity (1,000 pigs per four rooms).

## Simulations

All analyses and models were run in R version 4.1.0 [62] using the GillespieSSA [46] package and tidyverse [63] for plotting. All control measures were compared to the baseline model to determine their impact on the magnitude of total number of infected pigs per production cycle, probability of at least one workforce infection per production cycle, time from infected pig introduction to first workforce infection, and the number of stochastic extinctions (instances where the initially infected pig did not infect any other pigs or workers). Lastly, we compared different combinations of control measures. The baseline and intervention models were run for 5,000 and 10,000 iterations to confirm that the key results were stable and similar.

## Uncertainty analyses

To test the effects of aleatory uncertainty arising from the stochastic process implemented in our model, the coefficient of variation was calculated for total infected pigs and days to first workforce infection. To explore the impacts of variability and epistemic uncertainty, we performed an uncertainty analysis. For this uncertainty analysis, we implemented a deterministic

**Table 2. Incidence attributed to between-room transmission assuming workers work in a single direction from youngest (room 4) to oldest (room 1) pigs.**

| Room Numbers | Equations |
| --- | --- |
| 4 | $\beta_N S_4(0)$ |
| 3 | $\beta_N S_3(I_4)$ |
| 2 | $\beta_N S_2(I_3+I_4)$ |
| 1 | $\beta_N S_1(I_2+I_3+I_4)$ |

**Table 3. Simulated influenza transmission dynamics under separate control measures in pigs that did not have maternal derived antibodies.**

| Control Measure (CM) | Scenarios | Total Pigs Infected: Median [95% Uncertainty interval] (CV) | Probability of Workforce infection | Days from first infected pig to first Workforce Infection: Median [95% Uncertainty interval] (CV) | Probability of stochastic extinction |
|---|---|---|---|---|---|
| Baseline Model | No control measures | 3957 [0–3971] (0.332) | 0.617 | 20.9 [12.0–33.3] (0.104) | 0.0904 |
| Mass vaccination of pigs prior to arrival on farm. | VE = 20% | 3161 [0–3174] (0.377) | 0.5216 | 23.6 [13.2–36.1] (0.108) | 0.109 |
| | VE = 40% | 2362 [0–2374] (0.450) | 0.4098 | 28.4 [15.2–43.4] (0.123) | 0.1446 |
| | VE = 60% | 1540 [0–1559] (0.586) | 0.2654 | 38.0 [18.7–58.3] (0.153) | 0.209 |
| Isolation of infectious pigs up to a maximum of 10 pigs | 1/q = 3 days | 3950 [0–3967] (0.594) | 0.494 | 24.0 [13.8–36.4] (0.104) | 0.204 |
| | 1/q = 2 days | 3945 [0–3965] (0.730) | 0.4248 | 25.6 [14.9–38.7] (0.108) | 0.258 |
| | 1/q = 1 day | 2 [0–3960] (1.21) | 0.2704 | 29.9 [17.3–47.3] (0.127) | 0.382 |
| Directional Workforce Flow from Room 4 to Room 1 only. | Intro: Room 4 | 3956 [0–3971] (0.332) | 0.6176 | 20.8 [12.1–32.9] (0.104) | 0.0904 |
| | Intro: Room 3 | 2974 [0–3968] (0.350) | 0.5062 | 21.0 [11.5–33.5] (0.138) | 0.09 |
| | Intro: Room 2 | 1987 [0–2983] (0.396) | 0.3718 | 19.8 [10.8–33.8] (0.204) | 0.0906 |
| | Intro: Room 1 | 996 [0–1977] (0.449) | 0.2242 | 17.4 [10.3–27.6] (0.314) | 0.914 |

Results of 5000 iterations of each given scenario. Control Measure refers to the intervention implemented for those model scenarios. CV = Coefficient of Variation; VE = Vaccine efficacy; 1/q = mean time until an infectious pig is isolated; Intro = room to which infected pig is introduced. Total Pigs infected does not include the initially infected pig on day 42. Days from first infected was calculated using only iterations in which a workforce infection occurred and is the number of days to workforce infection after the initially infected pigs was introduced on day 42. Stochastic extinction was assumed to occur when the initially infected pig introduced on day 42 did not infect any workers or pigs during their infectious period.

approach using Monte Carlo sampling (5000 samples) of distributions of values for each of the model parameters (**Table 1**). We used deterministic models for evaluating the epistemic uncertainty in order to reduce computational effort. We compared results from the uncertainty analysis (deterministic models) to results of our stochastic models for single control measures only (**S1 Table**).

## Results

Our baseline model which assumed no maternal derived antibodies (MDAs), no vaccination, no isolation, no workforce routine changes showed that in 90.1% of iterations, almost the entire pig population (> 3,000 pigs) was infected after the introduction of single infected pig within the last room to be populated, on day 42 of the production cycle (**Tables 3 and 4**). Approximately 91% of the iterations resulted in transmission of influenza from the initially infected pig to another pig on the farm. The number of days (after the initially infected pig was introduced on day 42) until peak infection was reached was 19.5 days [95% uncertainty interval 0–23.1 days] (**Fig 2A**). The probability of at least one workforce infection per production cycle was 0.62 with the first workforce infection occurring on average within 21.5 days of infected pig introduction (median 20.9 [range 12.0–33.3]; 5,000 iterations).

In our model that assumed maternal derived antibodies, but no other influenza control measures only 37.3% of iterations indicated almost the entire pig population (> 3,000 pigs) being infected and 46% of iterations resulted in no transmission from the initially infected pig. Similarly, the probability of workforce infection was reduced to 0.25 and the median number of infected pigs reduced to 1 [0–3958] compared to 3957 [0–3971] (**Table 4**). The median time until peak infection (after the initially infected pigs was introduced on day 42) also reduced to 0 [0–40.6] (**Fig 2B**).

**Table 4. Simulated influenza transmission dynamics under separate control measures in pigs that did have maternal derived antibodies.**

| Control Measure (CM) | Scenarios | Total Pigs Infected: Median [95% Uncertainty interval] (CV) | Probability of Workforce infection | Days from first infected pig to first Workforce Infection: Median [95% Uncertainty interval] (CV) | Probability of stochastic extinction |
|---|---|---|---|---|---|
| Baseline Model | No control measures | 1 [0–3958] (1.30) | 0.2498 | 35.8 [19.9–49.9] (0.118) | 0.4634 |
| Mass vaccination of pigs prior to arrival on farm. | VE = 20% | 0 [0–3163] (1.56) | 0.176 | 39.2 [22.1–56.7] (0.126) | 0.5264 |
|  | VE = 40% | 0 [0–2364] (1.88) | 0.112 | 46.0 [27.6–62.4] (0.125) | 0.5824 |
|  | VE = 60% | 0 [0–1541] (2.38) | 0.0512 | 56.4 [32.2–76.1] (0.146) | 0.677 |
| Isolation of infectious pigs up to a maximum of 10 pigs | 1/q = 3 days | 0 [0–3929] (3.94) | 0.04 | 35.2 [20.6–52.8] (0.130) | 0.6994 |
|  | 1/q = 2 days | 0 [0 – 5] (5.19) | 0.021 | 32.5 [18.3–54.7] (0.144) | 0.752 |
|  | 1/q = 1 day | 0 [0 – 2] (11.3) | 0.006 | 32.7 [22.0–58.8] (0.147) | 0.8426 |
| Directional Workforce Flow from Room 4 to Room 1 only. | Intro: Room 4 | 1 [0–3954] (1.36) | 0.2298 | 37.2 [20.8–51.9] (0.121) | 0.4634 |
|  | Intro: Room 3 | 1 [0–2980] (1.32) | 0.203 | 39.2 [22.5–53.0] (0.138) | 0.4656 |
|  | Intro: Room 2 | 1 [0–1990] (1.45) | 0.138 | 39.9 [28.2–54.6] (0.150) | 0.4872 |
|  | Intro: Room 1 | 0 [0–994] (1.99) | 0.0578 | 36.7 [28.6–48.0] (0.190) | 0.5062 |

Results of 5000 iterations of each given scenario. Control Measure refers to the intervention implemented for those model scenarios. CV = Coefficient of Variation; VE = Vaccine efficacy; 1/q = mean time until an infectious pig is isolated; Intro = room to which infected pig is introduced. Total Pigs infected does not include the initially infected pig on day 42. Days from first infected was calculated using only iterations in which a workforce infection occurred and is the number of days to workforce infection after the initially infected pigs was introduced on day 42. Stochastic extinction was assumed to occur when the initially infected pig introduced on day 42 did not infect any workers or pigs during their infectious period.

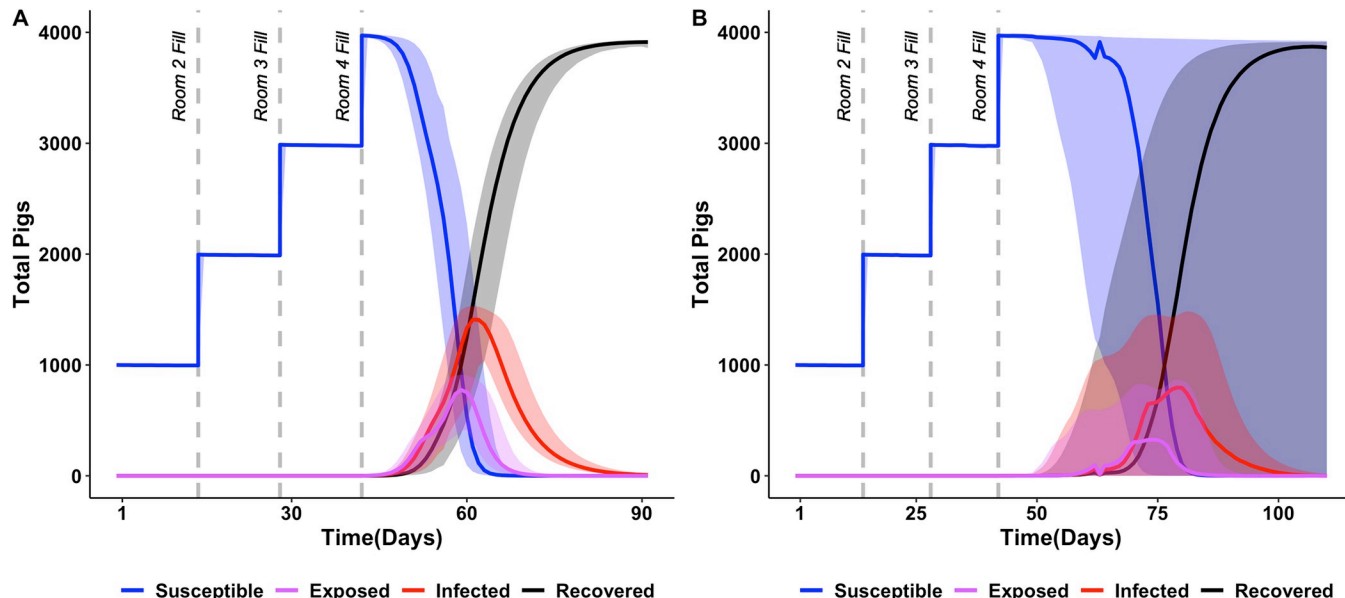

**Fig 2.** Influenza dynamics upon introduction of single infected hog on day 42 A) Transmission dynamics in pig populations without maternal-derived antibodies and B) Transmission dynamics in pig populations with maternal-derived antibodies. Baseline scenarios—unvaccinated pigs, assuming density dependent transmission. 4,000 total hogs on the farm in 4 rooms populated 2 weeks apart. Solid lines represent the median of 5,000 model iterations and shaded areas represent the 95% uncertainty interval.

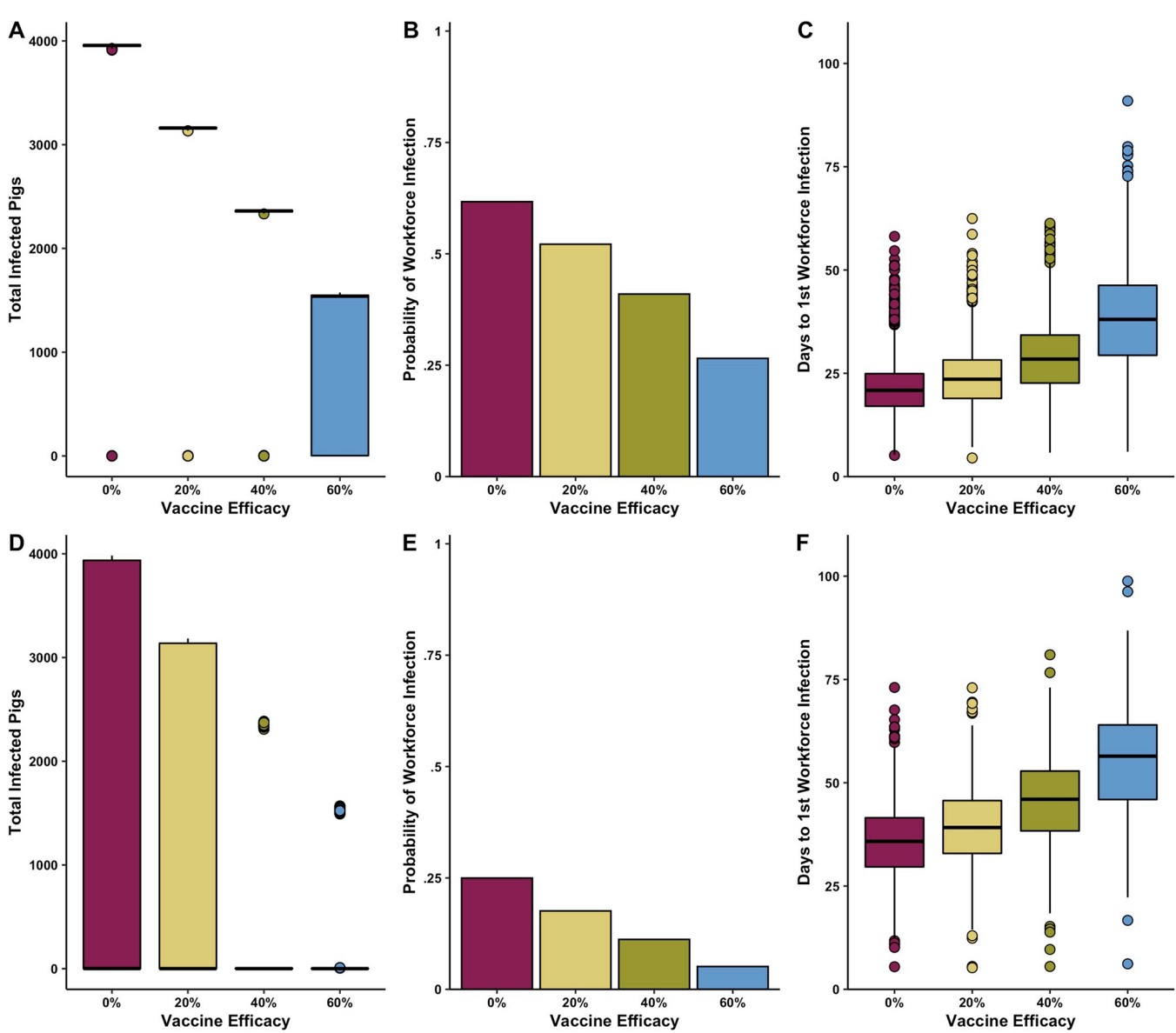

**Fig 3.** A)Total number of infected pigs without MDAs B)Probability of at least one workforce infection during one production cycle with mass vaccination of hog population and C) days from first infected pig introduction to first workforce infection with mass vaccination of hog population D)Total number of infected pigs with MDAs E)Probability of at least one workforce infection during one production cycle with mass vaccination of hog population and F) days from first infected pig introduction to first workforce infection with mass vaccination of hog population. Assumed introduction of a single infectious pig on day 42. density dependent transmission. 4,000 total hogs on the farm in 4 separate rooms.

Assuming that the swine vaccine that was 20% efficacious, we found that the median total number of infected pigs was reduced from 3957 [0–3971] to 3161 [1 – 3174] in pigs that did not have MDAs (**Table 3**) (**Fig 3A**). In pigs that were assumed to have MDAs a 20% efficacious vaccine showed less effect and reduced the total number of infected pigs to 0 [0–3163] form 1 [0–3958] (**Table 4**) (**Fig 3D**). A 40% efficacious vaccine reduced the total number of infected pigs without MDAs to 2362 [0–2374] and the total number of infected pigs with MDAs was reduced to 0 [0–2364]. A 60% efficacious vaccine showed the largest reduction in the total number of infected pigs with MDAs 1540 [0–1559] (**Tables 3 & 4**). The probability of workforce infection was reduced from 0.25 in the baseline model to 0.18, 0.11, and 0.05 for a 20%,

40% or 60% effective influenza vaccine, respectively (**Fig 3B and 3C**). A similar trend was seen for pigs with MDAs. The probability of workforce infection reduced from the baseline 0.2498 to 0.176, 0.112, and 0.0512 for a 20%, 40% or 60% effective influenza vaccine, respectively (**Fig 3E and 3F**).

When infected pigs without MDAs were isolated 1 day after they were infectious (q = 1) the model predicted a large reduction in the total number of infected pigs (2 [0–3960]) compared to the baseline model (3957 [0–3971]). In pigs that were assumed to have MDAs quarantining sick hogs within one day of becoming infectious also reduced the total number of infected pigs (0 [0 – 2]) compared to the MDA baseline model (1 [0–3958]). There was a minor decrease in the total number of infected pigs (3950 [0–3967] compared to the baseline model (3957 [0–3971]when quarantining pigs at a more realistic three days after becoming infectious (q = 0.33) (**Table 3 and Fig 4A**). In the pig populations where we assumed MDAs a similar minor reduction in the total number of infected pigs (0 [0–3929]) compared to the MDA base-line model (1 [0–3958]) was seen (**Table 4 and Fig 4D**). Quarantining infectious animals (in both the MDA and no-MDA scenarios) also reduced the probability of workforce infection and time to first workforce infection compared to baseline (**Tables 3 and 4, Fig 4B, 4C, 4E and 4F**).

Assuming workers move in a single direction from youngest pig batches to oldest pig batches led to a decreased median number of infected pigs in both MDA scenarios (**Tables 3 and 4**). When the infected pig was introduced in the youngest batch (Room 4), we found no significant change from the baseline scenario when comparing total infected pigs, probability of workforce infection, and time to first workforce infection (**Tables 3 and 4**). However, if the infected pig was introduced earlier into the simulation in one of the older batches (Room 1 or Room 2 for example), we found a reduction in the median total number of infected pigs (Room 1 introduction: 996 [0–1977]; Room 2 introduction: 1987 [0–2983]) compared to base-line (3957 [0–3971]) (**Fig 5A**). Our models also indicated that the probability of workforce infection (**Fig 5B**) was reduced and time to first workforce infection was increased when intro-ducing a directional workforce routine (**Fig 5C**). We found similar results in the model scenar-ios that assumed MDA within the pig populations (**Table 4, Fig 5D–5F**).

## Results of combination of control measures

When combining isolation of infected pigs (q = 0.33) with mass vaccination (with 40% Vac-cine Efficacy), the model predicted a reduction from baseline to 2340 [0–2362] total infected pigs without MDAs (**Table 5**). In pigs that were assumed to have MDAs, under the same con-trol strategy combination (q = 0.33, and 40% efficacious mass vaccination) we saw a reduction to 0 [0 – 3] total infected pigs (**Table 6**). Furthermore, the probability of a workforce infection was reduced from baseline to 0.25 per production cycle in which a single infected pig was introduced and no MDAs present. Assuming MDAs are present for incoming pigs, the proba-bility of workforce infection was reduced to 0.01. There was also an increase in time from infected pig introduction to first workforce infection for both scenarios.

The model predicted additional benefits when combining a directional work routine with mass vaccination using a 40% efficacious vaccine in pig populations without MDAs (**Table 5**). In the best-case scenario, when the infected pig is introduced with the oldest pigs, only 595 [0–598] total pigs were infected. This combination of interventions also reduced the probability of workforce infection to 0.12 and increased the time to first workforce infection to 21.5 [12.8–35.7] compared to the baseline model (probability of workforce infection = 0.25; time to first workforce infection = 35.8 days [19.9–49.9]). Similar trends were found when assuming incoming pigs had MDAs (Table 6).

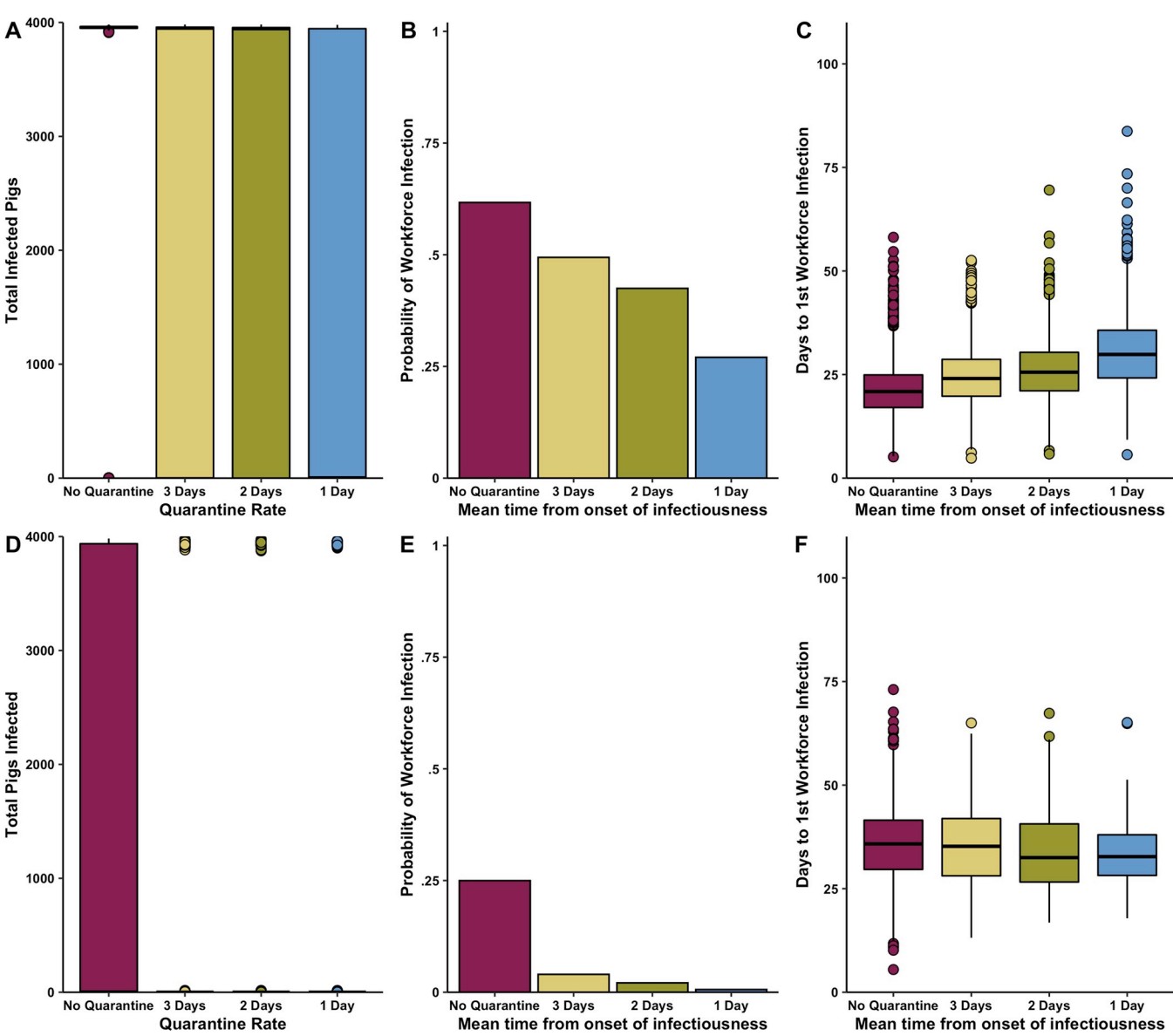

**Fig 4.** A)Total number of infected pigs without MDAs B) Probability of at least one workforce infection per production cycle with isolation of infectious hogs C) days from first infected pig introduction to first workforce infection with isolation of infectious hogs D)Total number of infected pigs with MDAs E) Probability of at least one workforce infection per production cycle with isolation of infectious hogs and F) days from first infected pig introduction to first workforce infection with isolation of infectious hogs. Assumed introduction of a single infectious pig on day 42, density dependent transmission and a maximum isolated population of 10 pigs per room. 4,000 total hogs on the farm in 4 separate rooms.

When all control measures were implemented (isolation of infected pigs, mass vaccination, and workforce operation) in pigs assumed to not have MDAs we found that the probability of workforce infection decreased to 0.02 per production cycle, even when the infected pig was introduced in the youngest batch of pigs (Room 4),with a 40% vaccine efficacy, and being able to isolate a sick pig within two days of being infectious (**Table 7**). In the model scenarios that assumed pigs had MDA protection for the first three weeks of being in the model we found similar results (Table 8). Notability, if the infected pig was introduced with the room 4 batch of pigs (on day 42), a 20% efficacious vaccine, and being able to isolate a sick pig within three days of being infectious the probability of a workforce infection was reduced to 0.01 (**Table 8**).

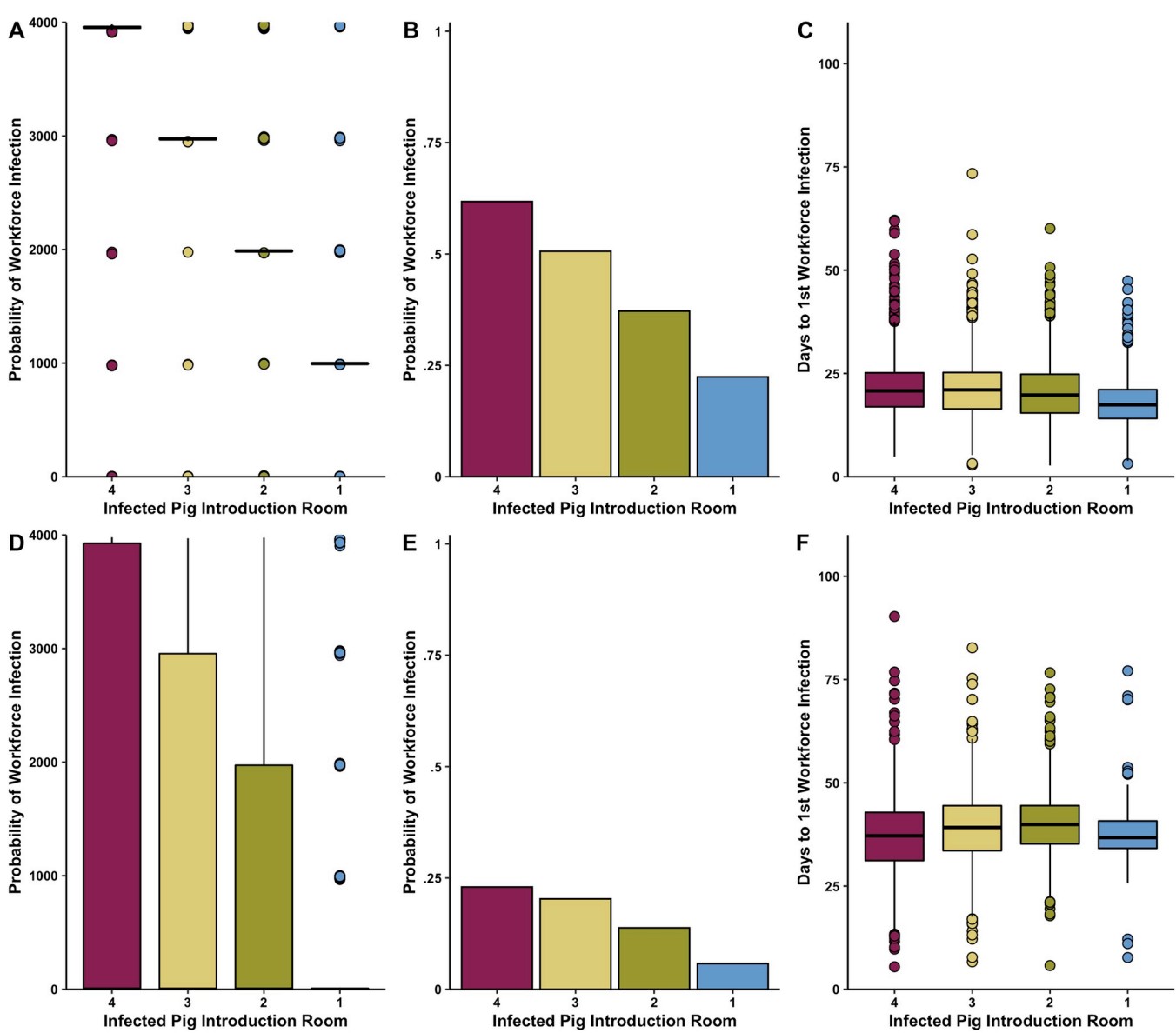

**Fig 5.** A)Total number of infected pigs without MDAs B) Probability of at least one workforce infection per production cycle with a directional workforce routine C) days from first infected pig introduction to first workforce infection with a directional workforce routine D)Total number of infected pigs without MDAs E) Probability of at least one workforce infection per production cycle with a directional workforce routine and F) days from first infected pig introduction to first workforce infection with a directional workforce routine. Assumed introduction of a single infectious pig on day 42, density dependent transmission and strict adherence of workforce working from younger pig batches to older batches. 4,000 total hogs on the farm in 4 separate rooms.

### Improved worker PPE

To test the effects of worker PPE and adherence to PEE mandates on influenza transmission from pigs to workers, simulations were performed using the same assumptions and parameters as the baseline model for pigs with MDAs. We found that when a worker wore a surgical mask the probability of pig-to-worker transmission was unchanged from 0.25. However, when workers adhered to strictly wearing an N95 mask, the probability of pig-to-worker transmission was reduced to 0.03. We also performed simulations to determine the impacts of a worker infected with influenza virus (swine associated H1N1, H1N2 or H3N2) infecting the pig

**Table 5. Simulated influenza transmission dynamics under combined control measures assuming no MDAs.**

| Vaccination Efficacy | Isolation Rate | Infected Pig Room Introduction | Median Total Pigs Infected [95% Uncertainty interval] (CV) | Probability of Workforce infection | Median Days to First Workforce Infection [95% Uncertainty interval] (CV) | Probability of stochastic extinction |
|---|---|---|---|---|---|---|
| 20% | 1/q = 3 days | Room 4 | 3151 [0–3169] (0.707) | 0.3716 | 28.3 [15.9–42.7] (0.115) | 0.2488 |
| | 1/q = 2 days | | 3144 [0–3166] (0.876) | 0.3254 | 30.2 [17.3–46.8] (0.123) | 0.305 |
| | 1/q = 1 day | | 1 [0–3159] (1.66) | 0.1516 | 36.7 [20.1–58.2] (0.147) | 0.4354 |
| 40% | 1/q = 3 days | Room 4 | 2340 [0–2362] (0.879) | 0.2516 | 36.2 [19.0–56.0] (0.142) | 0.3092 |
| | 1/q = 2 days | | 1 [3 – 2358] (1.18) | 0.1904 | 40.8 [20.8–62.1] (0.153) | 0.374 |
| | 1/q = 1 day | | 0 [0–2336] (3.23) | 0.0386 | 52.4 [26.5–78.9] (0.179) | 0.5066 |
| 60% | 1/q = 3 days | Room 4 | 1 [0–1490] (1.47) | 0.0836 | 58.7 [27.6–99.7] (0.221) | 0.4094 |
| | 1/q = 2 days | | 1 [0–1151] (2.80) | 2.20E-02 | 63.2 [27.8–109] (0.248) | 0.473 |
| | 1/q = 1 day | | 0 [0 – 12] (7.51) | 6.00E-04 | 28.2 [26.3–70.2] (0.324) | 0.5806 |
| **Workforce Flow Implemented** | | | | | | |
| 20% | 0 | Room 4 | 3161 [0–3174] (0.378) | 0.5232 | 23.6 [13.3–36.7] (0.109) | 0.109 |
| | | Room 3 | 2376 [0–3169] (0.394) | 0.4368 | 23.6 [12.6–38.2] (0.149) | 0.1094 |
| | | Room 2 | 1588 [0–2383] (0.430) | 0.3216 | 23.0 [12.3–35.8] (0.204) | 0.108 |
| | | Room 1 | 796 [0–799] (0.476) | 0.1796 | 19.5 [11.0–31.4] (0.324) | 0.1112 |
| 40% | 0 | Room 4 | 2362 [0–2374] (0.451) | 0.4164 | 28.4 [15.2–43.2] (0.123) | 0.1446 |
| | | Room 3 | 1775 [0–1787] (0.445) | 0.3334 | 27.8 [14.9–42.9] (0.158) | 0.1344 |
| | | Room 2 | 1186 [0–1195] (0.494) | 0.2332 | 25.9 [13.8–40.6] (0.208) | 0.1492 |
| | | Room 1 | 595 [0–598] (0.513) | 0.1204 | 21.5 [12.8–35.7] (0.308) | 0.1414 |
| 60% | 0 | Room 4 | 1538 [0–1558] (0.589) | 0.264 | 37.7 [19.3–60.1] (0.155) | 0.209 |
| | | Room 3 | 1157 [0–1174] (0.586) | 0.2072 | 37.2 [19.4–60.5] (0.199) | 0.2054 |
| | | Room 2 | 774 [0–785] (0.587) | 0.14 | 32.4 [17.1–57.7] (0.267) | 0.1946 |
| | | Room 1 | 388 [0–395] (0.629) | 0.0798 | 27.5 [15.7–46.0] (0.318) | 0.2016 |

Results of 5000 iterations of each given scenario. Control Measure refers to the intervention implemented for those model scenarios. CV = Coefficient of Variation; VE = Vaccine efficacy; 1/q = mean time until an infectious pig is isolated; Intro = room to which infected pig is introduced (Day 0 for Room 1, Day 14 for Room 2, Day 28 for Room 3 and Day 42 for Room 4). Total Pigs infected does not include the initially infected pig on day 42. Days from first infected was calculated using only iterations in which a workforce infection occurred and is the number of days to workforce infection after the initially infected pigs was introduced on day 42. Stochastic extinction was assumed to occur when the initially infected pig did not infect any workers or pigs during their infectious period.

population (with MDAs) on our growing unit. The simulations found that the probability of at least one pig being infected from an infected worker was 0.54. Our results also showed that this probability was reduced to 0.25 when workers wore a surgical mask and 0.08 when wearing an N95 mask, assuming the effective contact rate was reduced by the same percentage as published effectiveness values [60], and workers had perfect adherence to facial covering policies. We also found that the median number of pigs infected during these outbreaks was 1 [0–3966] for no mask, 0 [0–3958] when workers wore a surgical mask, and 0 [0–3934] when workers wore N95 masks. Results followed a similar pattern when we assumed that incoming pigs did not have MDAs. We found at baseline the probability of a worker being infected form a sick pig when not wearing any facial covering was 0.617. When it was that assumed workers wore surgical masks the probability of a worker being infected increased slightly to 0.6182 however, this probability was reduced to 0.088 when assuming workers wore N95 masks. The

**Table 6. Simulated influenza transmission dynamics under combined control measures assuming MDAs.**

| Vaccination Efficacy | Isolation Rate | Infected Pig Room Introduction | Median Total Pigs Infected [95% Uncertainty interval] (CV) | Probability of Workforce infection | Median Days to First Workforce Infection [95% Uncertainty interval] (CV) | Probability of stochastic extinction |
|---|---|---|---|---|---|---|
| 20% | 1/q = 3 days | Room 4 | 0 [0 – 7] (4.74) | 0.0252 | 37.3 [20.1–60.3] (0.160) | 0.744 |
| | 1/q = 2 days | | 0 [0 – 3] (6.63) | 0.0134 | 38.5 [25.2–57.2] (0.149) | 0.7948 |
| | 1/q = 1 day | | 0 [0 – 1] (17.8) | 0.0018 | 35.1 [21.5–57.6] (0.164) | 0.8754 |
| 40% | 1/q = 3 days | Room 4 | 0 [0 – 3] (6.59) | 0.0102 | 44.4 [30.5–74.1] (0.146) | 0.7842 |
| | 1/q = 2 days | | 0 [0 – 2] (9.00) | 0.0046 | 48.7 [25.5–77.8] (0.186) | 0.8362 |
| | 1/q = 1 day | | 0 [0 – 1] (25.4) | 0.0004 | 62.4 [56.1–68.7] (0.0948) | 0.892 |
| 60% | 1/q = 3 days | Room 4 | 0 [0 – 1] (6.85) | < 0.0002 | > 180 days | 0.9268 |
| | 1/q = 2 days | | 0 [0 – 1] (18.2) | 0.0002 | 60.0 [60.0–60.0] (NA) | 0.8812 |
| | 1/q = 1 day | | 0 [0 – 2] (10.1) | 0.003 | 65.8 [29.2–119] (0.291) | 0.849 |
| **Workforce Flow Implemented** | | | | | | |
| 20% | 0 | Room 4 | 0 [0–3158] (1.65) | 0.1578 | 40.2 [22.5–55.6] (0.124) | 0.5264 |
| | | Room 3 | 0 [0–2377] (1.47) | 0.1578 | 42.2 [27.4–56.7] (0.128) | 0.5212 |
| | | Room 2 | 0 [0–1588] (1.56) | 0.1136 | 42.5 [30.0–58.6] (0.161) | 0.536 |
| | | Room 1 | 0 [0–794] (2.17) | 0.0366 | 38.0 [30.1–49.9] (0.196) | 0.5568 |
| 40% | 0 | Room 4 | 0 [0–2358] (1.98) | 0.0974 | 46.1 [26.7–63.2] (0.127) | 0.5824 |
| | | Room 3 | 0 [0–1776] (1.59) | 0.111 | 48.2 [32.7–64.6] (0.134) | 0.5706 |
| | | Room 2 | 0 [0–1186] (1.69) | 0.0776 | 47.9 [32.1–67.8] (0.169) | 0.5776 |
| | | Room 1 | 0 [0–593] (2.54) | 0.0224 | 43.7 [32.0–61.9] (0.228) | 0.6126 |
| 60% | 0 | Room 4 | 0 [0–1533] (2.51) | 0.0498 | 56.2 [28.2–81.8] (0.158) | 0.677 |
| | | Room 3 | 0 [0–1159] (1.87) | 0.0628 | 57.8 [34.8–77.1] (0.156) | 0.6574 |
| | | Room 2 | 0 [0–775] (2.03) | 0.0356 | 58.3 [34.8–81.1] (0.203) | 0.6708 |
| | | Room 1 | 0 [0–387] (3.02) | 0.0098 | 52.2 [33.1–76.9] (0.263) | 0.7038 |

Results of 5000 iterations of each given scenario. Control Measure refers to the intervention implemented for those model scenarios. CV = Coefficient of Variation; VE = Vaccine efficacy; 1/q = mean time until an infectious pig is isolated; Intro = room to which infected pig is introduced (Day 0 for Room 1, Day 14 for Room 2, Day 28 for Room 3 and Day 42 for Room 4). Total Pigs infected does not include the initially infected pig on day 42. Days from first infected was calculated using only iterations in which a workforce infection occurred and is the number of days to workforce infection after the initially infected pigs was introduced on day 42. Stochastic extinction was assumed to occur when the initially infected pig did not infect any workers or pigs during their infectious period.

probability of a sick worker infecting a pig when not wearing any facial covering was 0.54 but improved to 0.24 and 0.076 when workers wore surgical masks and N95 masks respectively.

## Discussion

Our research aimed to simulate interspecies influenza transmission on a typical US indoor pig growing unit in order to test the effects of vaccination and non- pharmaceutical interventions, alone and in combination. To evaluate the effects of these interventions, we developed two baseline models: 1. where incoming pigs had no maternal derived antibodies (MDAs), and 2. where incoming pigs had MDAs. In these baseline models we assumed no vaccination was implemented at the growing farm, no pigs were isolated, and workers mixed randomly with the pigs. Upon introducing a single infected pig with a single subtype of influenza virus to population of pigs without MDAs, the model predicted an outbreak in 90.1% of our iterations that

**Table 7. Simulated influenza transmission dynamics when all control measures are implemented in a pig without MDAs.**

| Vaccination Efficacy | Isolation Rate | Infected Pig Room Introduction | Total Pigs Infected [95% Uncertainty interval] (CV) | Probability of Workforce infection | Days to first Workforce Infection [95% Uncertainty interval] (CV) | Probability of stochastic extinction |
|---|---|---|---|---|---|---|
| 20% | 1/q = 3 days | Room 4 | 3151 [0–3169] (0.709) | 0.3822 | 28.1 [16.4–43.3] (0.117) | 0.2488 |
| | | Room 3 | 2369 [0–2385] (0.700) | 0.3086 | 27.9 [16.1–41.9] (0.146) | 0.2452 |
| | | Room 2 | 1583 [0–1594] (0.757) | 0.2172 | 26.2 [14.6–41.8] (0.205) | 0.2574 |
| | | Room 1 | 793 [0–798] (0.778) | 0.1236 | 22.3 [13.5–35.1] (0.293) | 0.2532 |
| | 1/q = 2 days | Room 4 | 3143 [0–3166] (0.878) | 0.3174 | 30.7 [17.3–47.2] (0.125) | 0.305 |
| | | Room 3 | 2362 [0–2382] (0.896) | 0.2486 | 30.1 [17.4–46.2] (0.151) | 0.3056 |
| | | Room 2 | 1578 [0–1592] (0.926) | 0.1782 | 28.5 [16.4–45.0] (0.2) | 0.3074 |
| | | Room 1 | 791 [0–798] (0.966) | 0.097 | 24.3 [14.1–37.1] (0.306) | 0.3098 |
| | 1/q = 1 day | Room 4 | 1 [0–3159] (1.67) | 0.1474 | 36.9 [20.5–59.8] (0.144) | 0.4354 |
| | | Room 3 | 1 [0–2375] (1.69) | 0.1182 | 36.3 [20.6–57.9] (0.180) | 0.4396 |
| | | Room 2 | 1 [0–1587] (1.72) | 0.08 | 33.3 [17.7–56.7] (0.252) | 0.4298 |
| | | Room 1 | 1 [0–796] (1.70) | 0.0464 | 27.5 [17.1–50.5] (0.347) | 0.4264 |
| 40% | 1/q = 3 days | Room 4 | 2338 [0–2362] (0.885) | 0.2554 | 35.9 [19.5–54.9] (0.140) | 0.3092 |
| | | Room 3 | 1756 [0–1776] (0.901) | 0.183 | 35.3 [18.8–56.8] (0.184) | 0.3088 |
| | | Room 2 | 1172 [0–1188] (0.921) | 0.129 | 31.9 [18.1–52.5] (0.240) | 0.3058 |
| | | Room 1 | 588 [0–596] (0.961) | 0.0776 | 26.6 [16.1–43.6] (0.328) | 0.3116 |
| | 1/q = 2 days | Room 4 | 3 [0–2357] (1.19) | 0.1832 | 39.8 [20.9–62.4] (0.151) | 0.374 |
| | | Room 3 | 2 [0–1773] (1.22) | 0.1428 | 40.1 [20.4–64.2] (0.204) | 0.3766 |
| | | Room 2 | 3 [0–1185] (1.21) | 0.0946 | 36.1 [19.0–62.9] (0.272) | 0.365 |
| | | Room 1 | 3 [0–595] (1.25) | 0.0572 | 29.5 [18.4–49.3] (0.357) | 0.3656 |
| | 1/q = 1 day | Room 4 | 0 [0–2330] (3.33) | 0.0368 | 51.7 [28.3–85.6] (0.180) | 0.5066 |
| | | Room 3 | 0 [0–1755] (3.16) | 0.0324 | 47.9 [24.1–82.8] (0.221) | 0.506 |
| | | Room 2 | 1 [0–1173] (3.26) | 0.0162 | 38.2 [24.0–79.9] (0.284) | 0.4996 |
| | | Room 1 | 1 [0–590] (3.03) | 0.0104 | 34.7 [20.8–65.4] (0.359) | 0.4992 |
| 60% | 1/q = 3 days | Room 4 | 1 [0–1484] (1.49) | 0.0774 | 57.6 [28.2–96.7] (0.217) | 0.4094 |
| | | Room 3 | 2 [0–1120] (1.50) | 0.0546 | 55.4 [23.7–103] (0.285) | 0.406 |
| | | Room 2 | 1 [0–754] (1.52) | 0.0396 | 46.7 [24.3–95.5] (0.350) | 0.406 |
| | | Room 1 | 2 [0–381] (1.46) | 0.021 | 37.8 [21.6–65.0] (0.343) | 0.4008 |
| | 1/q = 2 days | Room 4 | 1 [0–1047] (2.81) | 0.0212 | 56.8 [31.8–106] (0.234) | 0.473 |
| | | Room 3 | 1 [0–796] (2.68) | 0.015 | 53.0 [26.2–92.6] (0.279) | 0.475 |
| | | Room 2 | 1 [0–707] (2.56) | 0.0152 | 55.0 [27.0–91.4] (0.310) | 0.4742 |
| | | Room 1 | 1 [0–374] (2.37) | 0.01 | 38.7 [21.9–81.9] (0.460) | 0.4676 |
| | 1/q = 1 day | Room 4 | 0 [0 – 12] (7.40) | 0.0008 | 28.4 [26.4–65.8] (0.277) | 0.5806 |
| | | Room 3 | 0 [0 – 10] (7.11) | <0.0002 | > 180 days | 0.5904 |
| | | Room 2 | 0 [0 – 11] (7.04) | <0.0002 | > 180 days | 0.5902 |
| | | Room 1 | 0 [0 – 10] (6.68) | 0.0002 | 45.5 [45.5–45.5] (NA) | 0.5956 |

Results of 5000 iterations of each given scenario. Control Measure refers to the intervention implemented for those model scenarios. CV = Coefficient of Variation; VE = Vaccine efficacy; 1/q = mean time until an infectious pig is isolated; Intro = room to which infected pig is introduced (Day 0 for Room 1, Day 14 for Room 2, Day 28 for Room 3 and Day 42 for Room 4). Total Pigs infected does not include the initially infected pig on day 42. Days from first infected was calculated using only iterations in which a workforce infection occurred and is the number of days to workforce infection after the initially infected pigs was introduced on day 42. Stochastic extinction was assumed to occur when the initially infected pig did not infect any workers or pigs during their infectious period.

infected almost the entire pig population (more than 3000 pigs infected) on the farm. This value was reduced to 37.3% when we assumed all incoming pigs had MDAs. Our findings were consistent with empirical data supporting widespread virus transmission in swine farms,

**Table 8. Simulated influenza transmission dynamics when all control measures are implemented in a pig without MDAs.**

| Vaccination Efficacy | Isolation Rate | Infected Pig Room Introduction | Total Pigs Infected [95% Uncertainty interval] (CV) | Probability of Workforce infection | Days to first Workforce Infection [95% Uncertainty interval] (CV) | Probability of stochastic extinction |
|---|---|---|---|---|---|---|
| 20% | 1/q = 3 days | Room 4 | 0 [0–6.05] (5.30) | 0.0144 | 33.5 [17.7–65.0] (0.181) | 0.744 |
| | | Room 3 | 0 [0 – 8] (4.87) | 0.0192 | 47.5 [18.9–65.6] (0.188) | 0.7562 |
| | | Room 2 | 0 [0 – 5] (6.08) | 0.0092 | 53.3[36.2–81.3] (0.197) | 0.7664 |
| | | Room 1 | 0 [0 – 2] (12.0) | 0.002 | 46.9 [43.7–73.3] (0.234) | 0.7726 |
| | 1/q = 2 days | Room 4 | 0 [0 – 3] (7.69) | 0.0062 | 34.6 [20.2–56.8] (0.153) | 0.7948 |
| | | Room 3 | 0 [0 – 3] (7.73) | 0.004 | 40.9 [23.2–65.8] (0.253) | 0.8062 |
| | | Room 2 | 0 [0 – 2] (9.61) | 0.0042 | 58.5 [31.9–84.7] (0.247) | 0.8134 |
| | | Room 1 | 0 [0 – 2] (17.6) | 0.0006 | 51.4 [44.4–60.4] (0.171) | 0.8164 |
| | 1/q = 1 day | Room 4 | 0 [0 – 1] (16.3) | 0.0006 | 28.6 [25.4–42.4] (0.135) | 0.8754 |
| | | Room 3 | 0 [0 – 1] (19.9) | 0.0006 | 50.4 [34.6–71.3] (0.255) | 0.877 |
| | | Room 2 | 0 [0 – 1] (38.5) | <0.0002 | > 180 days | 0.8824 |
| | | Room 1 | 0 [0 – 1] (3.14) | <0.0002 | > 180 days | 0.8796 |
| 40% | 1/q = 3 days | Room 4 | 0 [0 – 3] (7.13) | 0.0064 | 45.4 [27.4–70.7] (0.159) | 0.7842 |
| | | Room 3 | 0 [0 – 4] (6.57) | 0.0064 | 65.0 [32.3–82.0] (0.174) | 0.7976 |
| | | Room 2 | 0 [0 – 3] (7.18) | 0.005 | 64.8 [47.3–85.0] (0.176) | 0.8098 |
| | | Room 1 | 0 [0 – 2] (14.9) | 0.0002 | 93.1 [93.1–93.1] (NA) | 0.817 |
| | 1/q = 2 days | Room 4 | 0 [0 – 2] (10.3) | 0.0018 | 36.6 [18.1–57.2] (0.192) | 0.8362 |
| | | Room 3 | 0 [0 – 2] (10.9) | 0.0028 | 59.8 [30.0–80.6] (0.262) | 0.8442 |
| | | Room 2 | 0 [0 – 2] (13.5) | 0.0018 | 65.6[46.7–94.4] (0.222) | 0.8448 |
| | | Room 1 | 0 [0 – 1] (21.3) | <0.0002 | > 180 days | 0.857 |
| | 1/q = 1 day | Room 4 | 0 [0 – 1] (27.5) | 0.0002 | 44.5 [44.5–44.5] (NA) | 0.892 |
| | | Room 3 | 0 [0 – 1] (27.5) | 0.0004 | 54.7 [30.8–78.6] (0.454) | 0.899 |
| | | Room 2 | 0 [0 – 1] (31.5) | 0.0002 | 72.1 [721.1–72.1] (NA) | 0.9094 |
| | | Room 1 | 0 [0 – 1] (3.64) | <0.0002 | > 180 days | 0.9156 |
| 60% | 1/q = 3 days | Room 4 | 0 [0 – 2] (11.7) | 0.0008 | 93.4 [41.0–103] (0.268) | 0.849 |
| | | Room 3 | 0 [0 – 2] (10.5) | 0.001 | 84.7 [46.4–94.0] (0.224) | 0.855 |
| | | Room 2 | 0 [0 – 2] (11.6) | 0.0006 | 94.9 [49.9–112] (0.362) | 0.8618 |
| | | Room 1 | 0 [0 – 1] (18.2) | 0.0002 | 11.0 [11.0–11.0] (0) | 0.8674 |
| | 1/q = 2 days | Room 4 | 0 [0 – 1] (19.4) | 0.0002 | 57.7 [57.7–57.7] (NA) | 0.8812 |
| | | Room 3 | 0 [0 – 1] (19.3) | 0.0002 | 108 [108–108] (NA) | 0.886 |
| | | Room 2 | 0 [0 – 1] (20.9) | 0.0002 | 87.7 [87.7–87.7] (0) | 0.8904 |
| | | Room 1 | 0 [0 – 1] (24.3) | <0.0002 | > 180 days | 0.8962 |
| | 1/q = 1 day | Room 4 | 0 [0 – 1] (8.38) | <0.0002 | > 180 days | 0.9268 |
| | | Room 3 | 0 [0 – 1] (7.05) | <0.0002 | > 180 days | 0.9318 |
| | | Room 2 | 0 [0 – 1] (5.34) | <0.0002 | > 180 days | 0.934 |
| | | Room 1 | 0 [0 – 1] (4.09) | <0.0002 | > 180 days | 0.935 |

Results of 5000 iterations of each given scenario. Control Measure refers to the intervention implemented for those model scenarios. CV = Coefficient of Variation; VE = Vaccine efficacy; 1/q = mean time until an infectious pig is isolated; Intro = room to which infected pig is introduced (Day 0 for Room 1, Day 14 for Room 2, Day 28 for Room 3 and Day 42 for Room 4). Total Pigs infected does not include the initially infected pig on day 42. Days from first infected was calculated using only iterations in which a workforce infection occurred and is the number of days to workforce infection after the initially infected pigs was introduced on day 42. Stochastic extinction was assumed to occur when the initially infected pig did not infect any workers or pigs during their infectious period.

and consistent with modeling studies on different types of swine farms [39, 40, 42, 64]. However, the effective contact rate parameters in the model were based on relatively few empirical studies of different subtypes (all associated with swine). We did not simulate the transmission of subtypes typically associated with humans or avian species.

Assuming mass vaccination of swine with 40% efficacy, which is typical of vaccines currently in use, the median number of infected pigs was reduced to 2362 [0–2374] from 3957 [0–3971] in pigs assumed to not have MDAs, and 0 [0–23640. From 1 [0–3958) in pigs assumed to have MDAs. Furthermore, the probability of workforce infection was reduced by 0.14 to 0.11 in pigs with MDAs and by 0.21 (from 0.62 to 0.42) in pigs without MDAs. We explored a wide range of vaccine efficacy values to try and account for the annual variability in vaccine efficacy. Reducing the likelihood of transmission to the workforce can help reduce the likelihood of influenza of swine origin spreading to the local community, as well as ensuring a healthy workforce [65].

Overall, our results suggest that vaccination is beneficial in the control of endemic influenza, even with suboptimal efficacy. However, in cases of a novel influenza subtype, a vaccine may not be readily available. In these cases, non-pharmaceutical interventions were also predicted to be a modestly effective strategy to mitigate transmission. Our results indicated that quarantining symptomatic animals has some efficacy when applied in the absence of other control measures. These results imply, when compared to mass vaccination alone, being able to identify and completely isolate infected pigs early within their disease process can greatly reduce the total number of infected pigs and probability of worker being infected from a pig. In the face a novel influenza strain, implementing very strong surveillance efforts may help identify infected pigs before they are symptomatic and therefore help protect the herd. However, its impact was assumed to be limited due to limited space available to isolate infected pigs. Our predictions were also based on several other important assumptions: we assumed that isolated pens and biosecurity measures practiced were sufficient to prevent any onward transmission from isolated pigs to other pigs or humans, and that pigs were isolated on average 1, 2, or 3 days after the onset of infectiousness. While we were able to incorporate a complete isolation of these infected pigs in our model and to identify them in short timeframe this may not always be the case for every indoor hog farm and thus our results may overestimate the effectiveness of isolating infected pigs.

In some circumstances it may be more practical to implement a revised workforce routine, in which workers always work from the youngest pig batches first and move sequentially to the batches with older pigs. The model predicted up to 50% reduction in the number of infected pigs compared to the baseline models, depending on which the room the infected pig is introduced to. Further, adherence to the directional work routine greatly reduced the likelihood of the workforce being infected. This directional work routine could be further improved by workers entering rooms with known infected pigs last of all, each day. Our results also indicated that strict adherence to wearing an N95 mask greatly reduced the probability of workers becoming infected from sick pigs. This is not surprising as N95s are designed to protect the wearer from infection, and if fitted properly, they can reduce transmission as well, while surgical masks are designed to trap secretions from the wearer and therefore reduce transmission from that individual [61]. These results suggest in the face of a novel influenza subtype, strict adherence to new worker routines and wearing N95s will likely reduce the likelihood of pig-to-worker and worker-to-pig transmission.

We assumed that there were only two workers on a farm this size, which is typical of modern indoor hog production in the US. It is likely that human infections would be more common on farms with more workers, in which case modelling efforts should include worker-to-worker transmission. Further, influenza in swine peaks in spring and fall months, and therefore, probability of worker infection varies according to season, although it would be prudent to assume that the risk is present throughout the year [12].

We found that for most model simulations, the CV for total infected pigs and days to first workforce infection was less than one. However, our findings also suggest that more variability

was introduced (CV > 1) when we implemented multiple control measures in combination. Increasing the number of simulation iterations would likely lead to reduced CV values. Our epistemic uncertainty analysis (deterministic approach with Monte Carlo sampling for transmission parameters, latent and infectious periods) produced similar results (**S1 Table**) to our stochastic model except for timing of workforce infections (**Tables 3 and 4**). This discrepancy is due to the deterministic nature of the models used in this analysis. Results may also be an underestimation of the true probability of interspecies transmission and days to first workforce infection, as we derived our interspecies transmission parameters from a limited number of outbreak studies at outdoor agricultural fairs or research farms with a lower pig to worker ratio. Given the enclosed nature, large number and high density of pigs on an indoor pig growing unit, the interspecies transmission parameters we used in this study could be underestimated, although this may not be the case if high standards of biosecurity are adhered to. More empirical research into these transmission rates is needed.

We focused on transmission of a single subtype of influenza within a single production cycle on a single US indoor growing unit. We did not evaluate the likelihood of between production cycle infections that could arise from improper disinfection within swine housing facilities, equipment, or transportation vehicles. Future studies should evaluate the effects of control measures when multiple, heterologous IAV subtypes are introduced to indoor pig growing unit. Results from this study may not apply to outdoor units, facilities that are not grower specific, and swine practices outside of the US. We assumed no prior acquired immunity due to infection before pigs entered the growing facility. There were important knowledge gaps in the reviewed literature, which led to uncertainty in effective contact rate values. We based our transmission parameters from single experiments with a mixture of subtypes. Importantly, due to limited availability of empirical data the between-room transmission parameter combines both fomite and aerosol transmission. Interventions targeted at reducing aerosol transmission or fomite transmission may have very different impacts on between-room transmission, and these differences may also be sensitive to the farm building structure. However, despite these uncertainties our findings demonstrate the potential importance of influenza transmission between workers and pigs on US swine farms and demonstrate that relatively practical interventions could be applied to reduce transmission in order to mitigate the economic and public health impacts, especially in the face of a novel subtype. Further work is needed to test the acceptability of these measures to workers and swine farms, and to further understand the role that this swine-human interface has on the dynamics of influenza generally.

## Supporting information

**S1 Table. Epistemic uncertainty analysis results.** *to calculate days to first Workforce infection and the deterministic nature of these models we only used iterations in which to total infected workforce count exceeded 0.5.
(PDF)

## Author Contributions

**Conceptualization:** Eric Kontowicz, Max Moreno-Madriñan, Darryl Ragland, Wendy Beauvais.

**Data curation:** Eric Kontowicz.

**Formal analysis:** Eric Kontowicz.

**Funding acquisition:** Max Moreno-Madriñan, Darryl Ragland.

**Supervision:** Wendy Beauvais.

**Visualization:** Eric Kontowicz.

**Writing – original draft:** Eric Kontowicz.

**Writing – review & editing:** Eric Kontowicz, Max Moreno-Madriñan, Darryl Ragland, Wendy Beauvais.

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
