## [Decision Letter · Decision Letter 0]

25 Jan 2023

PONE-D-22-30000A stochastic compartmental model to simulate intra- and inter-species influenza transmission in an indoor swine farmPLOS ONE

Dear Dr. Kontowicz,

Thank you for submitting your manuscript to PLOS ONE. After careful consideration, we feel that it has merit but does not fully meet PLOS ONE’s publication criteria as it currently stands. Therefore, we invite you to submit a revised version of the manuscript that addresses the points raised during the review process.

We look forward to receiving your revised manuscript.

Kind regards,

Martial L Ndeffo-Mbah, Ph.D

Academic Editor

PLOS ONE

Journal Requirements:

Additional Editor Comments:

Thank you for you submission to PLoS ONE. Though our two reviewers found your manuscript interesting, they have raised major concerned about your modeling framework. These concerns are details in the reviewers reports. In revising your manuscript, please make sure to address all reviewers' concerns, as your revised manuscript will be sent back to reviewers for further evaluation before a final editorial decision can be made.

Reviewers' comments:

Reviewer's Responses to Questions

**Comments to the Author**

1. Is the manuscript technically sound, and do the data support the conclusions?

Reviewer #1: Partly

Reviewer #2: Yes

2. Has the statistical analysis been performed appropriately and rigorously? 

Reviewer #1: No

Reviewer #2: Yes

3. Have the authors made all data underlying the findings in their manuscript fully available?

Reviewer #1: Yes

Reviewer #2: Yes

4. Is the manuscript presented in an intelligible fashion and written in standard English?

Reviewer #1: Yes

Reviewer #2: Yes

5. Review Comments to the Author

Reviewer #1: The manuscript describes a mathematical model representing the spread of a influenza A virus in a typical US hog-growing unit. The originality of the model relies on the inclusion of a workforce compartment, allowing to evaluate the probability of pig-to-human transmission and vice-versa. Although the concept is really interesting, have some comments on the modelling framework, and statistical analyses could certainly help for outcomes interpretations

Introduction

L 50 - 12 Refs from 2 to 13 to say the risk of a pandemic is not-negligible is quite a lot.

L 53 – 55. Is it really necessary to come back to 1918 here?

L 56 Infuenza viruses are…

L 60. Influenza viruses are principally transmitted through the air. Not convinced environmental transmission by feces and urine is worth mentioning. Especially since the model does not account for the environment.

L 93. I would not be so strict on the protection conferred by MDAs. In fact, based on previous studies which evidenced MDAs as playing a role in IAV persistence, I was a bit surprised MDAs were not included in the model. I’ll come back to this later on.

L 111-118. Selected literature on IAV models is relatively sparse.

Material and Methods

Overview: Pitzer et al. described different types of farms, including an hog-growing production unit. White et al. represented a farrowing herd up to weaning. The question is why do the author state they adapted these models. Did they actually use and modify the code or did they develop their own model representing the population?

Population dynamics

The authors initialized the process with an empty farm, divided in four rooms which are sequentially filled in. How likely is such a situation? There might be a turnover to maintain the population relatively stable in real life? I’ll also come back on the workforce and the contact structure a bit later.

Transmission dynamics

The models are based on SEIR paradigm. In my opinion, maternally derived antibodies should be included in the modelling framework for the hog part, especially as the pigs are three weeks old when entering the herd.

The model is described at the room level for pigs, why aren’t there any indexes on the left-hand side of the equations.

How infection can protect from natural death?

Although I understand the logic, writing down differential equation for a population of 2 people is not appropriate.

For the chosen algorithm, I’m not convinced Gillespie is the best choice since select randomly one based on the relative contribution for each transition. It is clear that the probability of infection for pigs overcome the one of workers, which are in fact somehow independent. Using a tau-leap algorithm would allow to evaluate at each time step the probability of infections for both host populations. Of course the two populations interact but not in the same way. Another point for the workforce is the time spent in each room; that would really have been a plus. If the authors can evaluate the contribution of the workforce in the transmission according to the time spent in the rooms, it could help designing efficient walk-through. We can imagine workforces spending more time in post-weaning stage, when entering the herd.

Why did the authors use a density dependent formulation? The population in rooms are stable with the same density. No need to repeat the R0 formula three times.

The workers are in one room at a time, coming back to the above comment.

Chose to include or not mortality but please do not mix.

Interventions

The authors evaluated the vaccine efficacy corresponding to a proportion of pigs totally immune after vaccination. What if the vaccine reduce the transmission? Most of vaccines have the objective to limit clinical expression and thereby reduce viral shedding, but vaccinated animals can still be infected and transmit the virus.

When were the pigs vaccinated? If at three weeks of age, they might be some conflicts with maternal immunity?

For quarantine, the I_i equation has to be modified accordingly. The values of parameter q seem highly optimistic.

L241-242. The athors should indicate the percentage of reduction assumed with masks.

Simulations

How were the scenarios compared? Any statistical tests? For exemple survival analyses would be highly appreciated for the time to workforce infection. How was the stability of results appreciated?

Sensitivity analysis

Why deterministic models were used for the sensitivity analysis (Would rather call it an uncertainty analysis).

Results

The way results were presented looked not optimal to me. Texts around tables and figures could be more informative. Alternative presentations using statistical tests and/or survival analyses would help highlighting the key messages. The infection of workforce occurred after the peak in domestic pigs, refecting the comment on the algorithm. How the peak could be on day 42 (the day of introduction)?

In figure 2, the natural mortality does not seem to decrease the population. Did the authors run the deterministic version of there model?

The impact of vaccination is not clear. 40% of pigs were considered immune when integrating in the herd? The number of infections was therefore reduced by about 40%! The distributions of the number of infections could be interesting to see. In fact, the results with 80% protection looked more pronounced (with a median of only 23 pigs). However, we do not know the proportion of stochastic extinctions, with only one ore a few cases. When the transmission occurs, the number of infections should get close to 800? Here again, assuming a descrease of transmission by vaccinated animals whould have been a plus. Of course it complixifies the model but it wuld be more realistic.

The one day quarantine is really unrealistic (that’s mentioned in the discussion, but still). Where are located the quarantined pigs in real life? In a specific room (still potentially contributing to the transmission process between rooms) or a different building?

The impact of the introduction in the different rooms seems quite obvious: the spread only occur in the rooms that were previously filled in.

Owing to the previous comments, the combinations of the strategies are in line with expectations.

L 345.None of the simulations with 80% vaccine efficacy yield to workforce infection.

Improved worker PPE

Unless I missed it, the effect of PPE on transmission parameters were not specified.

Discussion

The discussion is well organized highlighting some key points of the manuscript. An example is the point on vaccination which was shown to reduce the disease spread. However, there is no word on maternally derived antibodies. Some points are also mentioned as the number of workforces which could be a crucial point especially with the Gillespie algorithm or the deterministic approaches used in this study. Finally, I would encourage the authors to account for the time spent in each rooms, which might be interesting for potential interventions.

Reviewer #2: The article “A stochastic compartmental model to simulate intra- and inter-species influenza 2 transmission in an indoor swine farm” seeks to elucidate the effectiveness of IAV control measures in a multispecies setting through the use of mechanistic modelling. This research is important because swine IAV is not only detrimental from an operational perspective but also poses human health risks. Though the zoonotic risk is low (as identified by the authors), unimpeded circulation of IAV in swine populations can lead to spillover events with pandemic potential, as witnessed in the 2009 IAV pandemic (see Hennig C. et al., 2022, Porc Health Manag). Unlike past models of swine IAV, the authors account for spillover events between swine and human hosts on virus circulation in a typical US production system. With this model, the effects of control strategies as they pertain to zoonotic transmission can now be assessed.

An SEIR population-based model was constructed and density-dependent transmission was assumed. Parameters were informed by the available literature. The results report the outcomes of multiple independent and combinatorial control strategies, using number of infected pigs, probability of a pig-to-worker spillover event, and time to a pig-to-worker spillover event. Both pharmaceutical and non-pharmaceutical interventions are assessed. In modelling control strategies, identification of infected animals was assumed to occur one or two days after the onset of infectiousness—assuming perfect detection. Though this may be an unrealistic assumption, the authors focused on modelling the overall effects of quarantine (composed of multiple components), and this highlights the rapidity of transmission of IAV within a swine herd. Sensitivity analysis was then conducted on relevant betas, sigmas, and gammas.

The results are cleanly presented, though figure 2 should be improved. Here, dashed lines are meant to indicate the 95% uncertainty interval however only the S and I compartments (blue and red lines, respectively) appear to have an interval (which confusingly the median (solid line) mostly falls outside of). The E and R compartments only have a single dashed line visible. If an interval is to be shown along with the median, a ribbon plot may be a better approach—though to avoid oversaturation of the plot it will likely have to be faceted by SEIR state. Otherwise, I recommend removing the dashed lines and solely present the median solid-line values, discussing the uncertainty in the text. Additionally, though the vertical dashed lines can be assumed to be the times of pen population, they should be explicitly labeled or removed.

A big hurdle of (many) modelling studies is parameterization, especially when using literature-derived parameters when there is a paucity of empirical data. The authors do mention these limitations, however specifically for the spillover parameters, further insight into the limitations of their derivation would be beneficial. Meaning, the hog-to-worker transmission rate was estimated from an outdoor agricultural fair and is now being used as a transmission rate at a densely-populated indoor facility. How does the change of environment affect the interpretation of this parameter (e.g. is this value expected to be a minimum or maximum rate in this new environment)? Additional discussion regarding workforce infections in the sensitivity analysis would help clarify this.

Overall, the paper is well written, the methodology is sound, and the results are interpreted in light of the limitations of the model. Further, this model avoids excess complexity, facilitating both comprehension and interpretation. Future extensions of this work could include accounting for imperfect detection, imperfect PPE adherence, multiple production cycles, and seasonality of IAV transmission, among many other directions. Lastly, I commend the authors on the cleanliness and readability of their R script.

Following these minor revisions, I recommend this article for publication

6. PLOS authors have the option to publish the peer review history of their article (what does this mean?). If published, this will include your full peer review and any attached files.

Reviewer #1: No

Reviewer #2: No

---

## [Author Response · Author response to Decision Letter 0]

28 Mar 2023

Dear Editor,

We would like to thank the editor and reviewers for their time, consideration, and comments on our manuscript. Below we have separated each reviewer’s comments and addressed them point-by-point. All line numbers included in our responses are in reference to the unmarked version of our revised paper. 

Reviewer #1: The manuscript describes a mathematical model representing the spread of a influenza A virus in a typical US hog-growing unit. The originality of the model relies on the inclusion of a workforce compartment, allowing to evaluate the probability of pig-to-human transmission and vice-versa. Although the concept is really interesting, have some comments on the modelling framework, and statistical analyses could certainly help for outcomes interpretations

Introduction

1. L 50 - 12 Refs from 2 to 13 to say the risk of a pandemic is not-negligible is quite a lot.

a. We have updated the references between lines 49 and 51. 

2. L 53 – 55. Is it really necessary to come back to 1918 here?

a. We appreciate the reviewer’s comment. We do think it is helpful to acknowledge the 1918 pandemic here, which is to date the largest and deadliest viral pandemic known. Additionally, there are influenza viruses circulating today that are genetically related to the 1918 pandemic strain. We have added language to the introduction to make this clearer. Lines 56-59.

3. L 56 Infuenza viruses are…

a. We have edited the manuscript to reflect this change (Line 60).

4. L 60. Influenza viruses are principally transmitted through the air. Not convinced environmental transmission by feces and urine is worth mentioning. Especially since the model does not account for the environment.

a. We have edited the manuscript to reflect this change.

5. L 93. I would not be so strict on the protection conferred by MDAs. In fact, based on previous studies which evidenced MDAs as playing a role in IAV persistence, I was a bit surprised MDAs were not included in the model. I’ll come back to this later on.

a. We appreciate the reviewer’s comments and agree that MDAs should be included in this research. To address this, we have extended the model structure and have completed model simulations that include maternally-derived antibodies (MDAs) and simulations that do not. We have updated our manuscript to include MDAs as follows:

i. We have included additional results using a reduced the direct transmission parameter (beta) for the first three weeks after the piglets arrive on the growing unit, to simulate the protective effect of MDAs in incoming piglets. After three weeks we assumed no more protection from MDAs and we increased the beta values to the baseline values.

ii. Results have been updated to reflect this change as well.

iii. The discussion has been expanded.

6. L 111-118. Selected literature on IAV models is relatively sparse.

a. We appreciate the reviewer’s comment on our selected literature. We understand that the IAV model literature is vast and broad. We choose to focus on the papers that directly influenced our model design and structure i.e. IAV transmission in swine production systems, which we feel we have summarized comprehensively. We have updated the language in the introduction to acknowledge the broad range of literature and indicate we only discussed the sources that directly influenced our work (Lines 121-123).

Material and Methods

7. Overview: Pitzer et al. described different types of farms, including an hog-growing production unit. White et al. represented a farrowing herd up to weaning. The question is why do the author state they adapted these models. Did they actually use and modify the code or did they develop their own model representing the population?

We have added text to the methods section to make it clear that we developed our own models using a similar compartmental structure to that of Pitzer et al and White et al. (Lines 133-137).

8. Population dynamics: The authors initialized the process with an empty farm, divided in four rooms which are sequentially filled in. How likely is such a situation? There might be a turnover to maintain the population relatively stable in real life? I’ll also come back on the workforce and the contact structure a bit later.

a. It is typical practice for indoor hog growing units in the Midwest USA (and elsewhere) to adopt an all-in all-out system for biosecurity reasons. Rooms are typically filled approximately two weeks apart as new batches of piglets reach the age of weaning . 4 rooms per farm is common. Our model is thus a very accurate representation of the production system. We added text in the methods and discussion to make this clearer (Lines 133).

Transmission dynamics

9. The models are based on SEIR paradigm. In my opinion, maternally derived antibodies should be included in the modelling framework for the hog part, especially as the pigs are three weeks old when entering the herd.

a. Thank you for this recommendation. Maternal derived antibodies have now been included into the model structure. This has been reflected in the methods, results, and discussion (Lines 160 – 161).

10. The model is described at the room level for pigs, why aren’t there any indexes on the left-hand side of the equations.

a. Thank you for this feedback. We have added subscript i’s to the left hand of the equations (Lines 179 – 188)

11. How infection can protect from natural death?

a. We have included death rates for infected pigs and made this clearer in the methods section (Lines186 & 217-218 and in Table 1).

12. Although I understand the logic, writing down differential equation for a population of 2 people is not appropriate.

a. We have removed the ODEs for the workforce but included text in the methods that discusses the workforce model structure (Lines 190 - 191. 

13. For the chosen algorithm, I’m not convinced Gillespie is the best choice since select randomly one based on the relative contribution for each transition. It is clear that the probability of infection for pigs overcome the one of workers, which are in fact somehow independent. Using a tau-leap algorithm would allow to evaluate at each time step the probability of infections for both host populations. Of course the two populations interact but not in the same way. 

a. We have clarified in the manuscript that we used the Gillespie Stochastic Simulation Algorithm Direct method (Lines 196 - 198). According to our understanding, the disadvantage of this method is simply that it is less efficient that other methods—such as the tau-leap method—which are approximations. We found that these approximations were not appropriate for our model (they gave unlikely results), and so we would argue that the Gillespie Stochastic Simulation Algorithm Direct method is the most appropriate in this case. 

14. Another point for the workforce is the time spent in each room; that would really have been a plus. If the authors can evaluate the contribution of the workforce in the transmission according to the time spent in the rooms, it could help designing efficient walk-through. We can imagine workforces spending more time in post-weaning stage, when entering the herd.

a. We appreciate and thank the reviewer for this comment, and we do agree that time spent in each room would likely play a role in the dynamics of interspecies transmission. However, currently, it is beyond the scope of the current research to evaluate this relationship with this level of granularity. Further, we cannot find evidence in the published literature or through expert consultation with swine farm owners that discusses what influence of a worker’s time on the probability of infection. As there is a low number of workers on indoor hog farms, the time in each room to complete daily tasks and routine health checks is unlikely to be modifiable by a significant amount. 

15. Why did the authors use a density dependent formulation? The population in rooms are stable with the same density. No need to repeat the R0 formula three times.

a. We assumed density dependent transmission is a better approximation than frequency dependent transmission for our model system; this is a typical assumption for respiratory agents. Intuitively, it seems reasonable to assume that if the population of pigs in a room increases in number, a single pig infected with influenza would infect more individuals, not a fixed number (as may be the case for a sexually transmitted infection for example). We have clarified this in the methods section of the manuscript. Lines 203-204.

b. The repeated beta calculations represent different point values of input parameters, due to uncertainty in the true parameter values that arises because of the limited experimental data in indoor environments that evaluated interspecies transmission. 

16. The workers are in one room at a time, coming back to the above comment.

a. We think that this comment is related to comment 14 and trying to estimate the amount of time a worker spends in a room and how that relates to the risk of a worker transmitting influenza to a pig. It is correct that workers are in one room at a time, however, it is extremely likely that each worker will go into each room that is housing pigs at least once per day. The timescale for our models was daily, and thus we incorporated workers as interacting with each room as such. Additionally, trying to estimate and minimize the amount of time each worker spends in each room is beyond the scope of this research at this time. 

17. Chose to include or not mortality but please do not mix.

a. We have updated our model and methods section to include mortality of infected pigs (Lines186 & 217-218 and in Table 1).

Interventions

The authors evaluated the vaccine efficacy corresponding to a proportion of pigs totally immune after vaccination. 

18. What if the vaccine reduce the transmission? Most of vaccines have the objective to limit clinical expression and thereby reduce viral shedding, but vaccinated animals can still be infected and transmit the virus.

a. We appreciate the reviewer’s comment highlighting the importance of distinguishing vaccine protection from clinical signs versus protection from infectiousness. In fact, we are not concerned with modelling clinical signs in our model, but rather interpret vaccine efficacy as the % reduction in transmission. Our parameters are based on experimental data that is appropriate for this estimation. We have clarified this this in the manuscript (Lines 221 – 224).

19. When were the pigs vaccinated? If at three weeks of age, they might be some conflicts with maternal immunity?

a. Since we have now amended our model structure to include protection conferred by maternally derived antibodies, we have also expanded the range of vaccine efficacy values to account for potential interaction between the protection conferred by the vaccine and maternally derived antibodies. New estimates are now 20%, 40%, and 60% as MDAs may interfere with vaccine efficacy and thus reduce it compared to the vaccine efficacy study data available, although there is a lack of published data to quantify the effect of MDAs on vaccine efficacy. It is common practice on indoor swine practices to vaccinate incoming pigs the day or two before they arrive at the indoor units (so at the breeder farms). 

b. We still chose to test a range of vaccine efficacy as these vaccines may have varying efficacy from year to year (Lines 481 – 483). Further, while we understand that MDAs can interfere with vaccine efficacy, we decided to keep the same efficacy values in our models that did not include MDAs. We chose to do this for ease of comparison and improved readability of results. 

c. We added to the methods section to make it clear that we assumed pigs were vaccinated the same day they were introduced to the farm (Lines 225 – 226).

20. For quarantine, the I_i equation has to be modified accordingly. The values of parameter q seem highly optimistic.

a. We have changed quarantine to isolation throughout the manuscript. We have expanded our testing to include isolation three days after infection (now shown in Table 1). We do understand that isolation within 1 to 2 days of infection may be highly optimistic, however we wanted to explore the range of impacts that this non-pharmaceutical intervention may have at reducing the impact of influenza on an indoor hog farm. 

21. L241-242. The authors should indicate the percentage of reduction assumed with masks.

a. We have included in our methods section that surgical masks were found to reduce the number of inhaled viral particles by 68% (1) and N95 mask by 91% (2). However, we also understand that % reduction in viral particles inhaled/exhaled is not the best proxy for % reduction in infection rate, however given the limited data we still choose to use this proxy (Lines 245 – 249).

Simulations

22. How were the scenarios compared? Any statistical tests? For exemple survival analyses would be highly appreciated for the time to workforce infection. How was the stability of results appreciated?

a. We did explore the differences between the scenarios using boxplots which compare the scenarios and also show the impact of stochasticity on our results (Figures 3 to 5). We also assessed the stability of results by repeating the stochastic simulations and plots multiple times and comparing the results to ensure qualitatively that they were similar. (This was added to the methods in lines 254-257) We are not convinced that a statistical test would add any credence to our findings and may give a misleading impression. For example, simply by increasing the number of iterations we would obtain a smaller p-value that would imply statistical significance. In reality, the absolute magnitude of the difference is what is important here. 

Sensitivity analysis

23. Why deterministic models were used for the sensitivity analysis (Would rather call it an uncertainty analysis).

a. we have updated our methods section to reflect that these are uncertainty analyses (Lines 261 – 270).

b. We chose to run deterministic models to evaluate epistemic uncertainty within our models due to computational and time limitations. To test the effects of epistemic uncertainty systematically through stochastic simulation would involve millions to billions of model iterations that evaluated different combinations and permutations of beta values (hog-to-hog, human-to-hog, and hog-to-human), pig latency and infectious periods, and human latency and infectious periods. Through a deterministic approach using Monte Carlo simulation to select random variables (within their give distributions) we could produce readable results and outcomes that could be compared to our stochastic models. 

Results

24. The way results were presented looked not optimal to me. Texts around tables and figures could be more informative. 

a. We have update table and figure legends to be more descriptive and informative.

25. Alternative presentations using statistical tests and/or survival analyses would help highlighting the key messages. The infection of workforce occurred after the peak in domestic pigs, refecting the comment on the algorithm. How the peak could be on day 42 (the day of introduction)?

a. We have updated the methods (lines 256 – 257) and results (lines 277 – 278) to discuss that timing of workforce infection is days since the initially infected pig was brought onto the farm. However, there were instances where the peak number of infected pigs was 1, and this occurs when there was no actual transmission from the initially infected pigs that is introduced on day 42 (in most scenarios) and thus in turn the day of peak infection will be 42. 

b. We also included text in table descriptions to clarify that days to first workforce infection is the number of days to first workforce infection after the initially infected pigs was introduced.

26. In figure 2, the natural mortality does not seem to decrease the population. Did the authors run the deterministic version of there model?

a. This was a stochastic model but the natural death rate is low. We have rerun all these models and figure 2 has been updated accordingly as well. 

27. The impact of vaccination is not clear. 40% of pigs were considered immune when integrating in the herd? The number of infections was therefore reduced by about 40%! The distributions of the number of infections could be interesting to see. In fact, the results with 80% protection looked more pronounced (with a median of only 23 pigs). However, we do not know the proportion of stochastic extinctions, with only one ore a few cases. When the transmission occurs, the number of infections should get close to 800? Here again, assuming a descrease of transmission by vaccinated animals whould have been a plus. Of course it complixifies the model but it wuld be more realistic.

a. We have added a new box plot to set of figures to show the total number of infected pigs to help highlight the variability within these interventions.

b. We have also updated the vaccine efficacy to take into account maternal antibodies. 

c. While we acknowledge that a partial reduction in transmissibility after vaccination is possible, even probable, we do not have access to data to justify a given reduction. Rather, we chose to use peer-reviewed published data that estimates the % reduction in pigs infected. But other studies have shown this is an appropriate approximation to population-level effects of vaccination (3). We have added to our results section the number of proportion of stochastic extinctions within the tables.

28. The one day quarantine is really unrealistic (that’s mentioned in the discussion, but still). Where are located the quarantined pigs in real life? In a specific room (still potentially contributing to the transmission process between rooms) or a different building?

a. We have added a sentence to our discussion section (lines 500 – 503) discussing that we may overestimate the impacts of quarantine as not every indoor hog farm may have the utility and resources to completely isolate these sick pigs and discover them in the same timely manner implemented in our model. We also highlight in this section though, that this non-pharmaceutical intervention if it is able to be implemented can still be extremely beneficial to the health of the pig herd when vaccines are not available but strong surveillance methods are (lines 491 – 495). 

29. The impact of the introduction in the different rooms seems quite obvious: the spread only occur in the rooms that were previously filled in.

a. We agree with the reviewer on this comment however we wanted to highlight this is the effects of the directional workforce movements. Further, as pointed out in the discussion section this can be implemented to ensure that workers work with potentially infected rooms last to help keep the remaining herd safer. This is not a current suggested best practice by the USDA or NPB.

30. Unless I missed it, the effect of PPE on transmission parameters were not specified.

a. We have added to the methods section how the mask effectiveness is related to a reduction in influenza transmission (Lines 245 – 249).

Discussion

31. The discussion is well organized highlighting some key points of the manuscript. An example is the point on vaccination which was shown to reduce the disease spread. However, there is no word on maternally derived antibodies. Some points are also mentioned as the number of workforces which could be a crucial point especially with the Gillespie algorithm or the deterministic approaches used in this study. 

a. We have updated the methods, results, and discussion with results from models that assumed maternal-derived antibody protection for the first three weeks of being on the simulated farm. 

32. Finally, I would encourage the authors to account for the time spent in each rooms, which might be interesting for potential interventions.

a. We understand that time spent in each room may have a large impact on the probability of worker being infected however, given the low number of workers on a indoor pig farm of this nature and the number of tasks a worker needs to complete each day, we felt it was beyond the scope of the current study to evaluate the timing in each room. Further, as these tasks need to be completed each day, it is unlikely that a worker could truly limit their time in each room in reality. However, it is more likely that we can ensure workers stick to a systematic approach of working with younger pigs first and moving form younger to older pigs. Or from known healthy pig rooms to suspected/confirmed infected pig rooms.

Reviewer #2: The article “A stochastic compartmental model to simulate intra- and inter-species influenza 2 transmission in an indoor swine farm” seeks to elucidate the effectiveness of IAV control measures in a multispecies setting through the use of mechanistic modelling. This research is important because swine IAV is not only detrimental from an operational perspective but also poses human health risks. Though the zoonotic risk is low (as identified by the authors), unimpeded circulation of IAV in swine populations can lead to spillover events with pandemic potential, as witnessed in the 2009 IAV pandemic (see Hennig C. et al., 2022, Porc Health Manag). Unlike past models of swine IAV, the authors account for spillover events between swine and human hosts on virus circulation in a typical US production system. With this model, the effects of control strategies as they pertain to zoonotic transmission can now be assessed.

An SEIR population-based model was constructed and density-dependent transmission was assumed. Parameters were informed by the available literature. The results report the outcomes of multiple independent and combinatorial control strategies, using number of infected pigs, probability of a pig-to-worker spillover event, and time to a pig-to-worker spillover event. Both pharmaceutical and non-pharmaceutical interventions are assessed. In modelling control strategies, identification of infected animals was assumed to occur one or two days after the onset of infectiousness—assuming perfect detection. Though this may be an unrealistic assumption, the authors focused on modelling the overall effects of quarantine (composed of multiple components), and this highlights the rapidity of transmission of IAV within a swine herd. Sensitivity analysis was then conducted on relevant betas, sigmas, and gammas.

1. The results are cleanly presented, though figure 2 should be improved. Here, dashed lines are meant to indicate the 95% uncertainty interval however only the S and I compartments (blue and red lines, respectively) appear to have an interval (which confusingly the median (solid line) mostly falls outside of). The E and R compartments only have a single dashed line visible. If an interval is to be shown along with the median, a ribbon plot may be a better approach—though to avoid oversaturation of the plot it will likely have to be faceted by SEIR state. Otherwise, I recommend removing the dashed lines and solely present the median solid-line values, discussing the uncertainty in the text. 

a. We have updated figure 2 to appropriately show the median and 95% uncertainty intervals. We also updated figure 2 to be ribbon plots instead of solid and dashed lines. 

2. Additionally, though the vertical dashed lines can be assumed to be the times of pen population, they should be explicitly labeled or removed.

a. Labels have been added to the gray dashed lines to indicate these are rooms being filled and to improve readability. 

3. A big hurdle of (many) modelling studies is parameterization, especially when using literature-derived parameters when there is a paucity of empirical data. The authors do mention these limitations, however specifically for the spillover parameters, further insight into the limitations of their derivation would be beneficial. Meaning, the hog-to-worker transmission rate was estimated from an outdoor agricultural fair and is now being used as a transmission rate at a densely-populated indoor facility. How does the change of environment affect the interpretation of this parameter (e.g. is this value expected to be a minimum or maximum rate in this new environment)? 

a. We have added text to the discussion section discussing this limitation and how we are likely underestimating this transmission potential in densely populated indoor pig farms of the US (Lines 530 – 538).

4. Additional discussion regarding workforce infections in the sensitivity analysis would help clarify this.

a. We have added to our discussion section a greater discussion regarding workforce infection in the sensitivity analysis (now named epistemic uncertainty analyses). In short, since deterministic models due not take chance into effect like stochastic models, workforce infection is guaranteed and thus will occur sooner compared to our stochastic models (Lines 530 – 538).

References Cited in our responses

1. Jefferson T, Foxlee R, Del Mar C, Dooley L, Ferroni E, Hewak B, et al. Physical interventions to interrupt or reduce the spread of respiratory viruses: systematic review. Bmj. 2008;336(7635):77-80.

2. EPA U, Network SSP. Pandemic influenza preparedness and response guidance for healthcare workers and healthcare employers. 2007.

3. Shim E, Galvani AP. Distinguishing vaccine efficacy and effectiveness. Vaccine. 2012;30(47):6700-5.

Our sincere thanks again for your comments and review of our manuscript.

---

## [Decision Letter · Decision Letter 1]

19 Apr 2023

A stochastic compartmental model to simulate intra- and inter-species influenza transmission in an indoor swine farm

PONE-D-22-30000R1

Dear Dr. Kontowicz,

We’re pleased to inform you that your manuscript has been judged scientifically suitable for publication and will be formally accepted for publication once it meets all outstanding technical requirements.

Kind regards,

Martial L Ndeffo-Mbah, Ph.D

Academic Editor

PLOS ONE

Additional Editor Comments (optional):

Reviewers' comments:

Reviewer's Responses to Questions

**Comments to the Author**

1. If the authors have adequately addressed your comments raised in a previous round of review and you feel that this manuscript is now acceptable for publication, you may indicate that here to bypass the “Comments to the Author” section, enter your conflict of interest statement in the “Confidential to Editor” section, and submit your "Accept" recommendation.

Reviewer #1: All comments have been addressed

Reviewer #2: All comments have been addressed

2. Is the manuscript technically sound, and do the data support the conclusions?

Reviewer #1: Yes

Reviewer #2: Yes

3. Has the statistical analysis been performed appropriately and rigorously? 

Reviewer #1: Yes

Reviewer #2: Yes

4. Have the authors made all data underlying the findings in their manuscript fully available?

Reviewer #1: Yes

Reviewer #2: Yes

5. Is the manuscript presented in an intelligible fashion and written in standard English?

Reviewer #1: Yes

Reviewer #2: Yes

6. Review Comments to the Author

Reviewer #1: The authors addressed each comment thoroughly with clarifications in the text. I thank the authors for the additional modelling effort on MDAs, which required a huge amount of work as highlighted by the code (which was made publicly available). Based on the fact the manuscript clearly presents all assumptions and the material given ensure reproducibility, I would now recommend to accept the manuscript. I nevertheless had fur minor comments (Line numbers refer to the manuscript with track changes):

L 134: could you check the sentence: “populations and speices toehr than indoor swine production systems”?

L154 : “included pigs with maternally derived antibodies,”: can this be considered as an “intervention strategy”. It depends on the status and vaccination strategy in the breeding herd.

L185: “where i is equal to 1, 2, 3 or 4, j represents the adjacent rooms where other hos pigs are housed, where j is equal to 1, 2, or 3”. It would be more correct to state j∈{1,2,3,4},j≠i . The same stands in the sums of the equations.

L318 I think we’re in the baseline scenario without MDAs here, the probability should be 0.61 according to table 2A.

L506 : “We found at baseline the probability of a worker being infected form a sick pig when not wearing any facial covering was 0.617 but improved to 0.6182 and 0.088 when a workers wore surgical masks or N95 respectively.” I don’t really see the improvement between 0.617 and 0.618?

Again, I want to thank the authors for the work done on the manuscript and the editor for giving me the opportunity to review the paper.

Reviewer #2: The authors have responded to all comments of reviewer 1 and 2 with major revisions having been made. This reviewer is satisfied with the revisions and recommends the article for publication.

7. PLOS authors have the option to publish the peer review history of their article (what does this mean?). If published, this will include your full peer review and any attached files.

Reviewer #1: No

Reviewer #2: No

---

## [Editor Report · Acceptance letter]

25 Apr 2023

PONE-D-22-30000R1 

A stochastic compartmental model to simulate intra- and inter-species influenza transmission in an indoor swine farm 

Dear Dr. Kontowicz:

I'm pleased to inform you that your manuscript has been deemed suitable for publication in PLOS ONE. Congratulations! Your manuscript is now with our production department. 

Kind regards, 

on behalf of

Dr. Martial L Ndeffo-Mbah 

Academic Editor

PLOS ONE